# Genomic reconstruction of the SARS-CoV-2 epidemic in England

Harald S. Vöhringer[1], Theo Sanderson[2,3], Matthew Sinnott[2], Nicola De Maio[1], Thuy Nguyen[2], Richard Goater[2], Frank Schwach[2,4], Ian Harrison[4], Joel Hellewell[5], Cristina V. Ariani[2], Sonia Gonçalves[2], David K. Jackson[2], Ian Johnston[2], Alexander W. Jung[1], Callum Saint[2], John Sillitoe[2], Maria Suciu[2], Nick Goldman[1], Jasmina Panovska-Griffiths[6], The Wellcome Sanger Institute COVID-19 Surveillance Team*, The COVID-19 Genomics UK (COG-UK) Consortium*, Ewan Birney[1], Erik Volz[7], Sebastian Funk[5], Dominic Kwiatkowski[2], Meera Chand[4,8], Inigo Martincorena[2], Jeffrey C. Barrett[2 ✉] & Moritz Gerstung[1,9 ✉]

The evolution of the severe acute respiratory syndrome coronavirus 2 (SARS-CoV-2) virus leads to new variants that warrant timely epidemiological characterization. Here we use the dense genomic surveillance data generated by the COVID-19 Genomics UK Consortium to reconstruct the dynamics of 71 different lineages in each of 315 English local authorities between September 2020 and June 2021. This analysis reveals a series of subepidemics that peaked in early autumn 2020, followed by a jump in transmissibility of the B.1.1.7/Alpha lineage. The Alpha variant grew when other lineages declined during the second national lockdown and regionally tiered restrictions between November and December 2020. A third more stringent national lockdown suppressed the Alpha variant and eliminated nearly all other lineages in early 2021. Yet a series of variants (most of which contained the spike E484K mutation) defied these trends and persisted at moderately increasing proportions. However, by accounting for sustained introductions, we found that the transmissibility of these variants is unlikely to have exceeded the transmissibility of the Alpha variant. Finally, B.1.617.2/Delta was repeatedly introduced in England and grew rapidly in early summer 2021, constituting approximately 98% of sampled SARS-CoV-2 genomes on 26 June 2021.

The SARS-CoV-2 virus accumulates approximately 24 point mutations per year, or 0.3 mutations per viral generation[1–3]. Most of these mutations appear to be evolutionarily neutral but, as the SARS-CoV-2 epidemic spread around the world during spring 2020, it became apparent that the virus is continuing to adapt to its human host. An initial sign was the emergence and global spread of the spike protein variant D614G in the second quarter of 2020. Epidemiological analyses estimated that this mutation, which defines the B.1 lineage, confers a 20% transmissibility advantage over the original A lineage that was isolated in Wuhan, China[4].

A broad range of lineages have been defined since that can be used to track SARS-CoV-2 transmission across the globe[5,6]. For example, B.1.177/EU-1 emerged in Spain in early summer 2020 and spread across Europe through travel[7]. Subsequently, four variants of concern (VOCs) have been identified by the WHO and other public health authorities: the B.1.351/Beta lineage was discovered in South Africa[8], where it spread rapidly in late 2020. The B.1.1.7/Alpha lineage was first observed in Kent in September 2020 (ref. [9]) from where it swept through the United Kingdom and large parts of the world due to a 50–60% increase[10–13] in transmissibility. P.1/Gamma originated in Brazil[14,15] and has spread throughout South America. Most recently, B.1.617.2/Delta was associated with a large surge of coronavirus disease 2019 (COVID-19) in India in April 2021 and subsequently around the world.

## Epidemiology of SARS-CoV-2 in England

In the United Kingdom, by late June 2021 the COVID-19 Genomics UK Consortium (COG-UK) had sequenced close to 600,000 viral samples. These data have enabled a detailed reconstruction of the dynamics of the first wave of the epidemic in the United Kingdom between February and August 2020 (ref. [16]). Here we leverage a subset of those data—genomic surveillance data generated at the Wellcome Sanger Institute—to characterize the growth rates and geographical spread of different SARS-CoV-2 lineages and reconstruct how newly emerging variants changed the course of the epidemic.

Our data cover England between 1 September 2020 and 26 June 2021, encompassing three epidemic waves and two national lockdowns (Fig. 1a). In this time period, we sequenced 281,178 viral genomes, corresponding to an average of 7.2% (281,178/3,894,234) of all of the positive tests from PCR testing for the wider population,

[1]European Molecular Biology Laboratory, European Bioinformatics Institute EMBL-EBI, Hinxton, UK. [2]Wellcome Sanger Institute, Hinxton, UK. [3]The Francis Crick Institute, London, UK. [4]Public Health England, London, UK. [5]London School of Hygiene & Tropical Medicine, London, UK. [6]The Big Data Institute, Nuffield Department of Medicine, University of Oxford, Oxford, UK. [7]MRC Centre for Global Infectious Disease Analysis, Jameel Institute for Disease and Emergency Analytics, Imperial College London, London, UK. [8]Guy's and St Thomas' NHS Foundation Trust, London, UK. [9]Division for AI in Oncology, German Cancer Research Centre DKFZ, Heidelberg, Germany. *Lists of authors and their affiliations appear online. ✉e-mail: jb26@sanger.ac.uk; moritz.gerstung@ebi.ac.uk

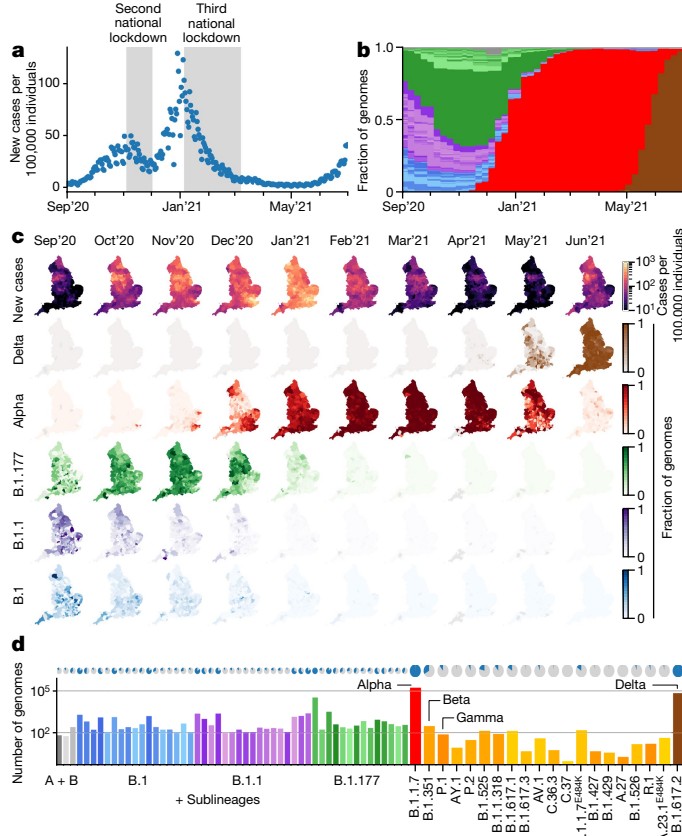

ranging from 5% in winter 2020 to 38% in early summer 2021, and filtered to remove cases that were associated with international travel (Methods and Extended Data Fig. 1a, b). Overall, a total of 328 SARS-CoV-2 lineages were identified using the PANGO lineage definition[5]. As some of these lineages were only rarely and intermittently detected, we collapsed these on the basis of the underlying phylogenetic tree into a set of 71 lineages for modelling (Fig. 1b–d and Supplementary Tables 1 and 2).

These data reveal a diversity of lineages in the fall of 2020 followed by sweeps of the Alpha and Delta variants (Fig. 1b and Supplementary Tables 2 and 3). Figure 1c shows the geographical distribution of cases and of different lineages, studied at the level of 315 English lower tier local authorities (LTLAs), administrative regions with approximately 100,000–200,000 inhabitants.

## Modelling the dynamics of SARS-CoV-2

We developed a Bayesian statistical model that tracks the fraction of genomes from different lineages in each LTLA in each week and fits the daily total number of positive Pillar 2 tests (Methods and Extended Data Fig. 2). The multivariate logistic regression model is conceptually similar to previous approaches in its estimation of relative growth rates[10,11]. It accounts for differences in the epidemiological dynamics between LTLAs, and enables the introduction of new lineages (Fig. 2a–c). Despite the sampling noise in a given week, the fitted proportions recapitulate the observed proportions of genomes as revealed by 35 example LTLAs covering the geography of England (Fig. 2b, c and Supplementary Notes 1 and 2). The quality of fit is confirmed by different probabilistic model selection criteria (Extended Data Fig. 3) and also evident at the aggregated regional level (Extended Data Fig. 4).

Although the relative growth rate of each lineage is modelled as identical across LTLAs, the local viral proportions change dynamically due to the timing and rate of introduction of different lineages. The model also calculates total and lineage-specific local incidences and time-dependent growth rates and approximate reproduction numbers $R_t$ by negative binomial spline fitting of the number of daily positive PCR tests (Methods, Fig. 2d and Extended Data Fig. 2c). Together, this enables a quantitative reconstruction of different periods of the epidemic, which we will discuss in chronological order.

**Fig. 1 | SARS-CoV-2 surveillance sequencing in England between September 2020 and June 2021. a**, Positive Pillar 2 SARS-CoV-2 tests in England. **b**, The relative frequency of 328 different PANGO lineages, representing approximately 7.2% of the tests shown in **a**. **c**, Positive tests (row 1) and the frequency of 4 major lineages (rows 2–5) across 315 English lower tier local authorities. **d**, The absolute frequency of sequenced genomes mapped to 71 PANGO lineages. The blue areas in the pie charts are proportional to the fraction of LTLAs in which a given lineage was observed.

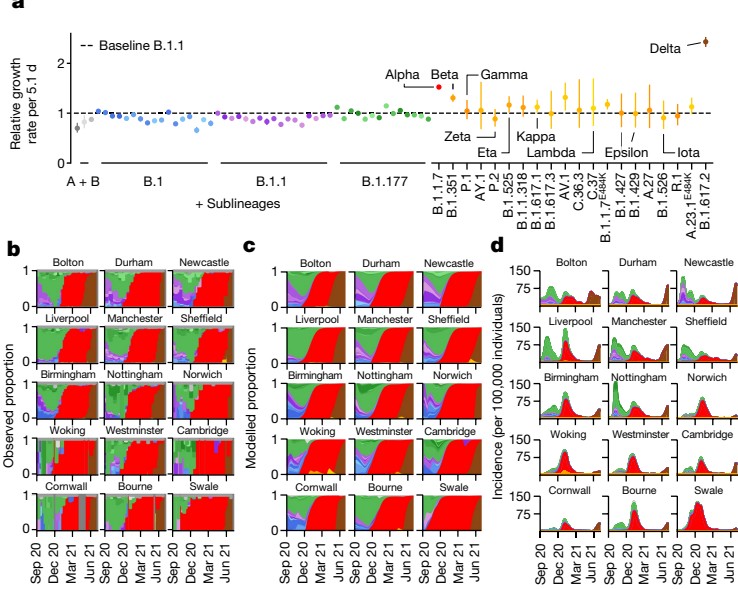

**Fig. 2 | Spatiotemporal model of 71 SARS-CoV-2 lineages in 315 English LTLAs between September 2020 and June 2021. a**, The average growth rates for 71 lineages. Data are median ± 95% CI. **b**, Lineage-specific relative frequency for 35 selected LTLAs, arranged by longitude and latitude to geographically cover England. **c**, Fitted lineage-specific relative frequency for the same LTLAs as in **b**. **d**, Fitted lineage-specific incidence for the same LTLAs as in **b**.

## Multiple subepidemics in autumn 2020

Autumn 2020 was characterized by a surge of cases—concentrated in the north of England—that peaked in November, triggering a second national lockdown (Fig. 1a, c). This second wave initially featured B.1 and B.1.1 sublineages, which were slightly more prevalent in the south and north of England, respectively (Fig. 2b, c). Yet, the proportion of B.1.177 and its geographically diverse sublineages steadily increased across LTLAs from around 25% at the beginning of September to 65% at the end of October. This corresponds to a growth rate of between 8% (growth per 5.1 d; 95% confidence interval (CI) = 7–9%) and 12% (95% CI = 11–13%) greater than that of B.1 or B.1.1. The trend of B.1.177 expansion relative to B.1 persisted throughout January (Extended Data Fig. 5a) and involved a number of monophyletic sublineages that arose in the UK, and similar patterns were observed in Denmark[17] (Extended Data Fig. 5b). Such behaviour cannot easily be explained by international travel, which was the major factor in the initial spread of B.1 throughout Europe in summer 2020 (ref. [7]). However, the underlying biological mechanism is unclear as the characteristic A222V spike variant is not believed to confer a growth advantage[7].

## The spread of Alpha during restrictions

The subsequent third wave from December 2020 to February 2021 was almost exclusively driven by Alpha/B.1.1.7, as described previously[10,11,18]. The rapid sweep of Alpha was due to an estimated transmissibility advantage of 1.52 compared with B.1.1 (growth per 5.1 d; 95% CI = 1.50–1.55; Fig. 2a), assuming an unchanged generation interval distribution[19]. The growth advantage is thought to stem, at least in part, from spike mutations that facilitate ACE2 receptor binding (N501Y)[20,21] and furin cleavage (P681H)[22]. Alpha grew during a period of restrictions, which proved to be insufficient to contain its spread (Fig. 3a).

The second national lockdown from 5 November to 1 December 2020 successfully reduced the total number of cases, but this masked a lineage-specific increase ($R_t > 1$; defined as growth per 5.1 d) in Alpha and a simultaneous decrease in other hitherto dominant lineages ($R_t < 1$) in 78% (246/315) of LTLAs[23] (Fig. 3b, c). This pattern of Alpha-specific growth during lockdown is supported by a model-agnostic analysis of raw case numbers and proportions of Alpha genomes (Fig. 3e).

Three levels of regionally tiered restrictions were introduced in December 2020 (ref. [24]) (Fig. 3a). The areas under different tiers of restrictions visibly and quantitatively coincide with the resulting local $R_t$ values, with greater $R_t$ values in areas with lower restrictions (Fig. 3a–c). The reopening caused a surge of cases across all tiers with $R_t > 1$, which is also evident in selected time series (Fig. 3d). As Alpha cases surged, more areas were placed under tier 3 restrictions, and stricter tier 4 restrictions were introduced. Nevertheless, Alpha continued to grow ($R_t > 1$) in most areas, presumably driven by increased social interaction over Christmas (Fig. 3c).

After the peak of 72,088 daily cases on 29 December 2020 (Fig. 1a), a third national lockdown was announced on 4 January 2021 (Fig. 3a). The lockdown and increasing immunity derived from infection and increasing vaccination[25] led to a sustained contraction of the epidemic to approximately 5,500 daily cases by 8 March, when restrictions began to be lifted by reopening schools (further steps of easing occurred on 12 April and 17 May). In contrast to the second national lockdown 93% (296/315) of LTLAs exhibited a contraction in both Alpha and other lineages (Fig. 3e).

## Elimination of lineages in early 2021

The lineage-specific rates of decline during the third national lockdown and throughout March 2021 resulted in large differences in lineage-specific incidence. Cases of Alpha contracted nationally from a peak of around 50,000 daily new cases to approximately 2,750 on

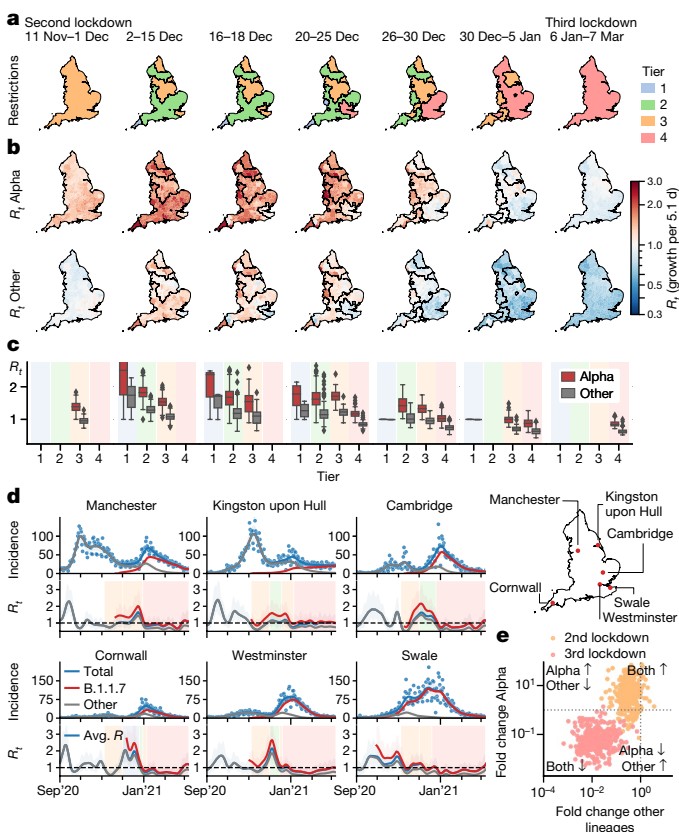

**Fig. 3 | Growth of B.1.1.7/Alpha and other lineages in relation to lockdown restrictions between November 2020 and March 2021. a**, Maps and dates of national and regional restrictions in England. Second national lockdown: closed hospitality businesses; contacts ≤ 2, outdoors only; open schools; reasonable excuse needed for leaving home[45]. Tier 1: private indoor gatherings of ≤6 persons. Tier 2: as tier 1 plus restricted hospitality services; gatherings of ≤6 in public outdoor places. Tier 3: as tier 2 plus most hospitality businesses closed. Tier 4: as tier 3 but single outdoor contact. Third national lockdown: closed schools with the exception of key workers. **b**, Local lineage-specific $R_t$ values for Alpha and the average $R_t$ value (growth per 5.1 d) of all of the other lineages in the same periods. **c**, $R_t$ values from $n = 315$ LTLA shown in **b**. The box centre horizontal line indicates the median, box limits show the quartiles, the whiskers extend to 1.5× the interquartile range. **d**, Total and lineage-specific incidence (top) and $R_t$ values (bottom) for six selected LTLAs during the period of restrictions. **e**, Crude lineage-specific fold changes (odds ratios) in Alpha and other lineages across the second (orange) and third national lockdown (red).

1 April 2021 (Fig. 4a). At the same time, B.1.177—the most prevalent lineage in November 2020—fell to less than an estimated 10 cases per day. Moreover, the incidence of most other lineages present in autumn 2020 was well below 1 after April 2021, implying that the majority of them have been eliminated. The number of observed distinct PANGO lineages declined from a peak of 137 to only 22 in the first week of April 2021 (Fig. 4b). Although this may be attributed in part to how PANGO lineages were defined, we note that the period of contraction did not replenish the genetic diversity lost due to the selective sweep by Alpha (Extended Data Fig. 6).

## Refractory variants with E484K mutations

Parallel to the elimination of many formerly dominant SARS-CoV-2 lineages, a number of new variants were imported or emerged (Fig. 4a). These include the VOCs B.1.351/Beta and P.1/Gamma, which carry the spike variant N501Y that is also found in B.1.1.7/Alpha and a similar pair of mutations (K417N/T and E484K) that were each shown to reduce the

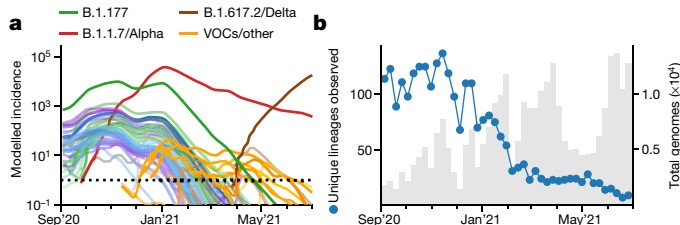

**Fig. 4 | Elimination of SARS-CoV-2 lineages during spring 2021. a**, Modelled lineage-specific incidence in England. The colours resemble major lineages as indicated and shades thereof indicate the respective sublineages. **b**, The observed number of PANGO lineages per week.

binding affinity of antibodies from vaccine-derived or convalescent sera[20,26–29]. The ability to escape from previous immunity is consistent with the epidemiology of Beta in South Africa[8] and especially the surge of Gamma in Manaus[15]. The variants B.1.525/Eta, B.1.526/Iota, B.1.1.318 and P.2/Zeta also harbour E484K spike mutations as per their lineage definition, and sublineages of Alpha and A.23.1 that acquired E484K were found in England (Fig. 5a, b).

The proportion of these E484K-containing variants was consistently 0.3–0.4% from January to early April 2021. A transient rise, especially of the Beta and Gamma variants, was observed in May 2021 (Fig. 5a, b). Yet, the dynamics were largely stochastic and characterized by a series of individual and localized outbreaks, possibly curtailed by local surge testing efforts against Beta and Gamma variants (Fig. 5c). Consistent with the transient nature of these outbreaks, the estimated growth rates of these variants were typically lower than Alpha (Fig. 2a).

Sustained imports from international travel were a critical driving mechanism behind the observed number of non-Alpha cases. A phylogeographical analysis establishing the most parsimonious sets of monophyletic and exclusively domestic clades, which can be interpreted as individual introductions, confirmed that A.23.1 with E484K (1 clade) probably has a domestic origin as no genomes of the same clade were observed internationally (Methods, Fig. 5d and Extended Data

Fig. 7). The estimated number of introductions was lowest for B.1.1.318 (3 introductions, range = 1–6), and highest for Beta (49 introductions, range = 45–58) and Eta (30 introductions, range = 18–34). Although our data exclude genomes sampled directly from travellers, these repeated introductions show that the true rate of transmission is lower than the observed increase in the number of surveillance genomes.

## The rise of Delta from April to June 2021

The B.1.617.1/Kappa and B.1.617.2/Delta lineages, which were first detected in India in 2020, first appeared in English surveillance samples in March 2021. In contrast to other VOCs, Delta/Kappa do not contain N501Y or E484K mutations, but their L452R mutation may reduce antibody recognition[27] and P681R enhances furin cleavage[30], similar to the P681H mutation of Alpha. The frequency of Delta, which harbours further spike mutations of unknown function, increased rapidly and reached levels of 98% (12,474/12,689) on 26 June 2021 (Fig. 5a, b). Although initially constrained to a small number of large local clusters, such as in Bolton, in May 2021 (Fig. 5c), Delta was detected in all LTLAs by 26 June 2021 (Fig. 1c). The sweep of Delta occurred at a rate of around 59% (growth per 5.1 d, CI = 53–66) higher than Alpha with minor regional variation (Fig. 2a, Extended Data Fig. 4e and Supplementary Table 4).

The rapid rise of Delta contrasts with Kappa, which grew more slowly despite being introduced at a similar time and into a similar demographic background (Figs. 2a and 5b). This is also evident in the phylogeographical analysis (based on data as of 1 May 2021). The 224 genomes of Delta derive from larger clades (23 introductions, range = 6–40; around 10 genomes for every introduction) compared with the 80 genomes of Kappa (17 introductions, range = 15–31; around 3–4 genomes per introduction) and also other variants (Fig. 5d and Extended Data Fig. 8). The AY.1 lineage, derived from Delta and containing an additional K417N mutation, appeared only transiently (Fig. 5b).

The sustained domestic growth of Delta and its international spread[31] relative to the Alpha lineage are the first evidence of a biological growth advantage. The causes appear to be a combination of increased transmissibility and immune evasion. Evidence for higher transmissibility

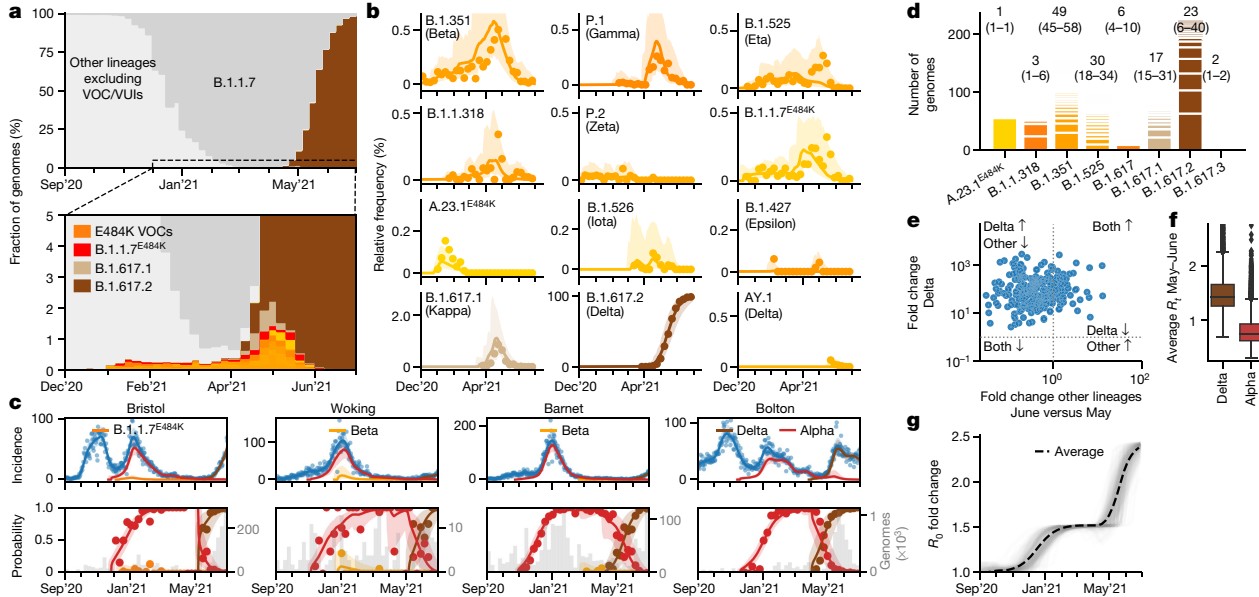

**Fig. 5 | Dynamics of E484K variants and Delta between January and June 2021. a**, The observed relative frequency of other lineages (light grey), Alpha/B.1.1.7 (dark grey), E484K variants (orange) and Delta/B.1.617.2 (brown). **b**, The observed and modelled relative frequency of variants in England. **c**, The total and relative lineage-specific incidence in four selected LTLAs. For **b** and **c**, the shaded areas indicate the 95% CIs. **d**, Estimated UK clade numbers (numbers in square

parentheses represent minimum and maximum numbers) and sizes. **e**, Crude growth rates (odds ratios) of Delta and Alpha between April and June 2021, as in Fig. 3e. **f**, Lineage-specific $R_t$ values of $n$ = 315 LTLA in the same period, defined as in Fig. 3c. **g**, Changes in the average transmissibility across 315 LTLAs during the study period.

includes the fast growth in younger unvaccinated age groups, reports of elevated secondary attack rates[32] and a higher viral load[33]. Furthermore, vaccine efficacy against infection by Delta is diminished, depending on the type of vaccine[34,35], and reinfection is more frequent[36], both supported by experimental research demonstrating the reduced antibody neutralization of Delta by vaccine-derived and convalescent sera[37,38].

The higher growth rate of Delta—combined with gradual reopening and proceeding vaccination—repeated the dichotomous pattern of lineage-specific decline and growth, although now with declining Alpha ($R_t < 1$) and growing Delta ($R_t > 1$; Fig. 5e, f). Overall, we estimate that the spread of more transmissible variants between August 2020 and early summer 2021 increased the average growth rate of circulating SARS-CoV-2 in England by a factor of 2.39 (95% CI = 2.25–2.42; Fig. 5g). Thus, previously effective interventions may prove to be insufficient to contain newly emerging and more transmissible variants.

## Discussion

Our dense genomic surveillance analysis identified lineages that consistently grew faster than others in each local authority and, therefore, at the same time, under the same restrictions and in a comparable population. This pinpointed a series of variants with elevated transmissibility, in broad agreement with other reports[10,11,13,15,31]. However, a number of limitations exist. The growth rates of rare new variants are stochastic due to introductions and superspreading. Local outbreaks of the Beta and Gamma variants triggered asymptomatic surge testing, which may have reduced their spread. Furthermore, transmission depends both on the viral variant and the immunity of the host population, which changed from less than 20% to over 90% in the study period[39]. This will influence the growth rates of variants with immune evasion capabilities over time. The effect of immunity is currently not modelled, but may become more important in the future as SARS-CoV-2 becomes endemic. Further limitations are discussed in the Limitations section of the Methods.

The third and fourth waves in England were each caused by more transmissible variants, which outgrew restrictions that were sufficient to suppress previous variants. During the second national lockdown, Alpha grew despite falling numbers for other lineages and, similarly, Delta took hold in April and May when cases of Alpha were declining. The fact that such growth was initially masked by the falling cases of dominant lineages highlights the need for dense genomic surveillance and rapid analysis to devise optimal and timely control strategies. Such surveillance should ideally be global as, even though Delta was associated with a large wave of cases in India, its transmissibility remained unclear at the time due to a lack of systematic genomic surveillance data.

The 2.4-fold increase in growth rate during the study period as a result of new variants is also likely to have consequences for the future course of the pandemic. If this increase in growth rate was explained solely by higher transmissibility, it would raise the basic reproduction number $R_0$ from a value of around 2.5–3 in spring 2020 (ref. [40]) to the range of 6–7 for Delta. This is likely to spur new waves of the epidemic in countries that have to date been able to control the epidemic despite low vaccination rates, and it may exacerbate the situation elsewhere. Although the exact herd-immunity threshold depends on contact patterns and the distribution of immunity across age groups[41,42], it is worth considering that Delta may increase the threshold to values around 0.85. Given current estimates of vaccine efficacy[34,35,43] this would require nearly 100% vaccination coverage. Even though more than 90% of adults had antibodies against SARS-CoV-2 (ref. [39]) and close to 70% had received two doses of vaccination, England saw rising Delta variant cases in the first weeks of July 2021. It can therefore be expected that other countries with high vaccination coverage are also likely to experience rising cases when restrictions are lifted.

SARS-CoV-2 is likely to continue its evolutionary adaptation process to humans[44]. To date, variants with considerably higher transmissibility

have had strongest positive selection, and swept through England during the 10 months of this investigation. However, the possibility that an increasingly immune population may now select for variants with better immune escape highlights the need for continued systematic and, ideally, global genomic surveillance.

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

**The Wellcome Sanger Institute COVID-19 Surveillance Team**

Irina Abnizova[2], Louise Aigrain[2], Alex Alderton[2], Mozam Ali[2], Laura Allen[2], Roberto Amato[2], Ralph Anderson[2], Cristina Ariani[2], Siobhan Austin-Guest[2], Sendu Bala[2], Jeffrey Barrett[2], Andrew Bassett[2], Kristina Battleday[2], James Beal[2], Mathew Beale[2], Charlotte Beaver[2], Sam Bellany[2], Tristram Bellerby[2], Katie Bellis[2], Duncan Berger[2], Matt Berriman[2], Emma Betteridge[2], Paul Bevan[2], Simon Binley[2], Jason Bishop[2], Kirsty Blackburn[2], James Bonfield[2], Nick Boughton[2], Sam Bowker[2], Timothy Brendler-Spaeth[2], Iraad Bronner[2], Tanya Brooklyn[2], Sarah Kay Buddenborg[2], Robert Bush[2], Catarina Caetano[2], Alex Cagan[2], Nicola Carter[2], Joanna Cartwright[2], Tiago Carvalho Monteiro[2], Liz Chapman[2], Tracey-Jane Chillingworth[2], Peter Clapham[2], Richard Clark[2], Adrian Clarke[2], Catriona Clarke[2], Daryl Cole[2], Elizabeth Cook[2], Maria Coppola[2], Linda Cornell[2], Clare Cornwell[2], Craig Corton[2], Abby Crackett[2], Alison Cranage[2], Harriet Craven[2], Sarah Craw[2], Mark Crawford[2], Tim Cutts[2], Monika Dabrowska[2], Matt Davies[2], Robert Davies[2], Joseph Dawson[2], Callum Day[2], Aiden Densem[2], Thomas Dibling[2], Cat Dockree[2], David Dodd[2], Sunil Dogga[2], Matthew Dorman[2], Gordon Dougan[2], Martin Dougherty[2], Alexander Dove[2], Lucy Drummond[2], Eleanor Drury[2], Monika Dudek[2], Jillian Durham[2], Laura Durrant[2], Elizabeth Easthope[2], Sabine Eckert[2], Pete Ellis[2], Ben Farr[2], Michael Fenton[2], Marcella Ferrero[2], Neil Flack[2], Howerd Fordham[2], Grace Forsythe[2], Luke Foulser[2], Matt Francis[2], Audrey Fraser[2], Adam Freeman[2], Anastasia Galvin[2], Maria Garcia-Casado[2], Alex Gedny[2], Sophia Girgis[2], James Glover[2], Sonia Goncalves[2], Scott Goodwin[2], Oliver Gould[2], Marina Gourtovaia[2], Andy Gray[2], Emma Gray[2], Coline Griffiths[2], Yong Gu[2], Florence Guerin[2], Will Hamilton[2], Hannah Hanks[2], Ewan Harrison[2], Alexandria Harrott[2], Edward Harry[2], Julia Harvison[2], Paul Heath[2], Anastasia Hernandez-Koutoucheva[2], Rhiannon Hobbs[2], Dave Holland[2], Sarah Holmes[2], Gary Hornett[2], Nicholas Hough[2], Liz Huckle[2], Lena Hughes-Hallet[2], Adam Hunter[2], Stephen Inglis[2], Sameena Iqbal[2], Adam Jackson[2], David Jackson[2], Keith James[2], Dorota Jamrozy[2], Carlos Jimenez Verdejo[2], Ian Johnston[2], Matthew Jones[2], Kalyan Kallepally[2], Leanne Kane[2], Keely Kay[2], Sally Kay[2], Jon Keatley[2], Alan Keith[2], Alison King[2], Lucy Kitchin[2], Matt Kleanthous[2], Martina Klimekova[2], Petra Korlevic[2], Ksenia Krasheninnkova[2], Dominic Kwiatkowski[2], Greg Lane[2], Cordelia Langford[2], Adam Laverack[2], Katharine Law[2], Mara Lawniczak[2], Stefanie Lensing[2], Steven Leonard[2], Laura Letchford[2], Kevin Lewis[2], Amanah Lewis-Wade[2], Jennifer Liddle[2], Quan Lin[2], Sarah Lindsay[2], Sally Linsdell[2], Rich Livett[2], Stephanie Lo[2], Rhona Long[2], Jamie Lovell[2], Jon Lovell[2], Catherine Ludden[2], James Mack[2], Mark Maddison[2], Aleksei Makunin[2], Irfan Mamun[2], Jenny Mansfield[2], Neil Marriott[2], Matt Martin[2], Inigo Martincorena[2], Matthew Mayho[2], Shane McCarthy[2], Jo McClintock[2], Samantha McGuigan[2], Sandra McHugh[2], Liz McMinn[2], Carl Meadows[2], Emily Mobley[2], Robin Moll[2], Maria Morra[2], Leanne Morrow[2], Kathryn Murie[2], Sian Nash[2], Claire Nathwani[2], Plamena Naydenova[2], Alexandra Neaverson[2], Rachel Nelson[2], Ed Nerou[2], Jon Nicholson[2], Tabea Nimz[2], Guillaume G. Noell[2], Sarah O'Meara[2], Valeriu Ohan[2], Karen Oliver[2], Charles Olney[2], Doug Ormond[2], Agnes Oszlanczi[2], Steve Palmer[2], Yoke Fei Pang[2], Barbora Pardubska[2], Naomi Park[2], Aaron Parmar[2], Gaurang Patel[2], Minal Patel[2], Maggie Payne[2], Sharon Peacock[10], Arabella Petersen[2], Deborah Plowman[2], Tom Preston[2], Liam Prestwood[2], Christoph Puethe[2], Michael Quail[2], Diana Rajan[2], Shavanthi Rajatileka[2], Richard Rance[2], Suzannah Rawlings[2], Nicholas Redshaw[2], Joe Reynolds[2], Mark Reynolds[2], Simon Rice[2], Matt Richardson[2], Connor Roberts[2], Katrina Robinson[2], Melanie Robinson[2], David Robinson[2], Hazel Rogers[2], Eduardo Martin Rojo[2], Daljit Roopra[2], Mark Rose[2], Luke Rudd[2], Ramin Sadri[10], Nicholas Salmon[2], David Saul[2], Frank Schwach[2], Carol Scott[2], Phil Seekings[2], Lesley Shirley[2], John Sillitoe[2], Alison Simms[2], Matthew Sinnott[2], Shanthi Sivadasan[2], Bart Siwek[2], Dale Sizer[2], Kenneth Skeldon[2], Jason Skelton[2], Joanna Slater-Tunstill[2], Lisa Sloper[2], Nathalie Smerdon[2], Chris Smith[2], Christen Smith[2], James Smith[2], Katie Smith[2], Michelle Smith[2], Sean Smith[2], Tina Smith[2], Leighton Sneade[2], Carmen Diaz Soria[2], Catarina Sousa[10], Emily Souster[2], Andrew Sparkes[2], Michael Spencer-Chapman[2], Janet Squares[2], Robert Stanley[2], Claire Steed[2], Tim Stickland[2], Ian Still[2], Michael R. Stratton[2], Michelle Strickland[2], Allen Swann[2], Agnieszka Swiatkowska[2], Neil Sycamore[2], Emma Swift[2], Edward Symons[2], Suzanne Szluha[2], Emma Taluy[2], Nunu Tao[2], Katy Taylor[2], Sam Taylor[2], Stacey Thompson[2], Mark Thompson[2], Mark Thomson[2], Nicholas Thomson[2], Scott Thurston[2], Gerry Tonkin-Hill[2], Dee Toombs[2], Benjamin Topping[2], Jaime Tovar-Corona[2], Daniel Ungureanu[2], James Uphill[2], Jana Urbanova[2], Philip Jansen Van Vuuren[2], Valerie Vancollie[2], Paul Voak[2], Danielle Walker[2], Matthew Walker[2], Matt Waller[2], Gary Ward[2], Charlie Weatherhogg[2], Niki Webb[2], Danni Weldon[2], Alan Wells[2], Eloise Wells[2], Luke Westwood[2], Theo Whipp[2], Thomas Whiteley[2], Georgia Whitton[2], Andrew Whitwham[2], Sara Widaa[2], Mia Williams[2], Mark Wilson[2] & Sean Wright[2]

[10]Department of Medicine, University of Cambridge, Cambridge, UK.

**The COVID-19 Genomics UK (COG-UK) Consortium**

**Funding acquisition, leadership and supervision, metadata curation, project administration, samples and logistics, sequencing and analysis, software and analysis tools, and visualization**
Samuel C. Robson[11,12]

**Funding acquisition, leadership and supervision, metadata curation, project administration, samples and logistics, sequencing and analysis, and software and analysis tools**
Thomas R. Connor[13,14] & Nicholas J. Loman[15]

**Leadership and supervision, metadata curation, project administration, samples and logistics, sequencing and analysis, software and analysis tools, and visualization**
Tanya Golubchik[6]

**Funding acquisition, leadership and supervision, metadata curation, samples and logistics, sequencing and analysis, and visualization**
Rocio T. Martinez Nunez[16]

**Funding acquisition, leadership and supervision, project administration, samples and logistics, sequencing and analysis, and software and analysis tools**
David Bonsall[6]

**Funding acquisition, leadership and supervision, project administration, sequencing and analysis, software and analysis tools, and visualization**
Andrew Rambaut[17]

**Funding acquisition, metadata curation, project administration, samples and logistics, sequencing and analysis, and software and analysis tools**
Luke B. Snell[18]

**Leadership and supervision, metadata curation, project administration, samples and logistics, software and analysis tools, and visualization**
Rich Livett[2]

**Funding acquisition, leadership and supervision, metadata curation, project administration, and samples and logistics**
Catherine Ludden[4,10]

**Funding acquisition, leadership and supervision, metadata curation, samples and logistics, and sequencing and analysis**
Sally Corden[14] & Eleni Nastouli[19,20,21]

**Funding acquisition, leadership and supervision, metadata curation, sequencing and analysis, and software and analysis tools**
Gaia Nebbia[18]

**Funding acquisition, leadership and supervision, project administration, samples and logistics, and sequencing and analysis**
Ian Johnston[2]

**Leadership and supervision, metadata curation, project administration, samples and logistics, and sequencing and analysis**
Katrina Lythgoe[6], M. Estee Torok[10,22] & Ian G. Goodfellow[23]

**Leadership and supervision, metadata curation, project administration, samples and logistics, and visualization**
Jacqui A. Prieto[24,25] & Kordo Saeed[24,26]

**Leadership and supervision, metadata curation, project administration, sequencing and analysis, and software and analysis tools**
David K. Jackson[2]

**Leadership and supervision, metadata curation, samples and logistics, sequencing and analysis, and visualization**
Catherine Houlihan[19,27]

**Leadership and supervision, metadata curation, sequencing and analysis, software and analysis tools, and visualization**
Dan Frampton[20,27]

**Metadata curation, project administration, samples and logistics, sequencing and analysis, and software and analysis tools**
William L. Hamilton[22] & Adam A. Witney[29]

**Funding acquisition, samples and logistics, sequencing and analysis, and visualization**
Giselda Bucca[28]

**Funding acquisition, leadership and supervision, metadata curation and project administration**
Cassie F. Pope[29,30]

**Funding acquisition, leadership and supervision, metadata curation, and samples and logistics**
Catherine Moore[14]

**Funding acquisition, leadership and supervision, metadata curation, and sequencing and analysis**
Emma C. Thomson[31]

**Funding acquisition, leadership and supervision, project administration, and samples and logistics**
Ewan M. Harrison[2,32]

**Funding acquisition, leadership and supervision, sequencing and analysis, and visualization**
Colin P. Smith[28]

**Leadership and supervision, metadata curation, project administration, and sequencing and analysis**
Fiona Rogan[33]

**Leadership and supervision, metadata curation, project administration, and samples and logistics**
Shaun M. Beckwith[34], Abigail Murray[34], Dawn Singleton[34], Kirstine Eastick[35], Liz A. Sheridan[36], Paul Randell[37], Leigh M. Jackson[38], Cristina V. Ariani[2] & Sónia Gonçalves[2]

**Leadership and supervision, metadata curation, samples and logistics, and sequencing and analysis**
Derek J. Fairley[33,39], Matthew W. Loose[40] & Joanne Watkins[14]

**Leadership and supervision, metadata curation, samples and logistics, and visualization**
Samuel Moses[41,42]

**Leadership and supervision, metadata curation, sequencing and analysis, and software and analysis tools**
Sam Nicholls[15], Matthew Bull[14] & Roberto Amato[2]

**Leadership and supervision, project administration, samples and logistics, and sequencing and analysis**
Darren L. Smith[43,44,45]

**Leadership and supervision, sequencing and analysis, software and analysis tools, and visualization**
David M. Aanensen[2,46] & Jeffrey C. Barrett[2]

**Metadata curation, project administration, samples and logistics, and sequencing and analysis**
Dinesh Aggarwal[2,4,10], James G. Shepherd[31], Martin D. Curran[47] & Surendra Parmar[47]

**Metadata curation, project administration, sequencing and analysis, and software and analysis tools**
Matthew D. Parker[48]

**Metadata curation, samples and logistics, sequencing and analysis, and software and analysis tools**
Catryn Williams[14]

**Metadata curation, samples and logistics, sequencing and analysis, and visualization**
Sharon Glaysher[49]

**Metadata curation, sequencing and analysis, software and analysis tools, and visualization**
Anthony P. Underwood[2,46], Matthew Bashton[43,44], Nicole Pacchiarini[14], Katie F. Loveson[12] & Matthew Byott[19,20]

**Project administration, sequencing and analysis, software and analysis tools, and visualization**
Alessandro M. Carabelli[10]

**Funding acquisition, leadership and supervision, and metadata curation**
Kate E. Templeton[17,50]

**Funding acquisition, leadership and supervision, and project administration**
Thushan I. de Silva[48], Dennis Wang[48], Cordelia F. Langford[2] & John Sillitoe[2]

**Funding acquisition, leadership and supervision, and samples and logistics**
Rory N. Gunson[51]

**Funding acquisition, leadership and supervision, and sequencing and analysis**
Simon Cottrell[14], Justin O'Grady[52,53] & Dominic Kwiatkowski[2,54]

**Leadership and supervision, metadata curation and project administration**
Patrick J. Lillie[35]

**Leadership and supervision, metadata curation, and samples and logistics**
Nicholas Cortes[55], Nathan Moore[55], Claire Thomas[55], Phillipa J. Burns[35], Tabitha W. Mahungu[56] & Steven Liggett[57]

**Leadership and supervision, metadata curation, and sequencing and analysis**
Angela H. Beckett[11,58] & Matthew T. G. Holden[59]

**Leadership and supervision, project administration, and samples and logistics**
Lisa J. Levett[60], Husam Osman[4,61] & Mohammed O. Hassan-Ibrahim[37]

**Leadership and supervision, project administration, and sequencing and analysis**
David A. Simpson[33]

**Leadership and supervision, samples and logistics, and sequencing and analysis**
Meera Chand[4], Ravi K. Gupta[32], Alistair C. Darby[62] & Steve Paterson[62]

**Leadership and supervision, sequencing and analysis, and software and analysis tools**
Oliver G. Pybus[63], Erik Volz[7], Daniela de Angelis[64], David L. Robertson[31], Andrew J. Page[52] & Inigo Martincorena[2]

**Leadership and supervision, sequencing and analysis, and visualization**
Louise Aigrain[2] & Andrew R. Bassett[2]

**Metadata curation, project administration, and samples and logistics**
Nick Wong[65], Yusri Taha[66], Michelle J. Erkiert[37] & Michael H. Spencer Chapman[2,32]

**Metadata curation, project administration, and sequencing and analysis**
Rebecca Dewar[50] & Martin P. McHugh[50,67]

**Metadata curation, project administration, and software and analysis tools**
Siddharth Mookerjee[68,69]

**Metadata curation, project administration and visualization**
Stephen Aplin[24], Matthew Harvey[24], Thea Sass[24], Helen Umpleby[24] & Helen Wheeler[24]

**Metadata curation, samples and logistics, and sequencing and analysis**
James P. McKenna[39], Ben Warne[70], Joshua F. Taylor[71], Yasmin Chaudhry[23], Rhys Izuagbe[23], Aminu S. Jahun[23], Gregory R. Young[43,44], Claire McMurray[15], Clare M. McCann[44,45], Andrew Nelson[44,45] & Scott Elliott[49]

**Metadata curation, samples and logistics, and visualization**
Hannah Lowe[41]

**Metadata curation, sequencing and analysis, and software and analysis tools**
Anna Price[13], Matthew R. Crown[44], Sara Rey[14], Sunando Roy[19] & Ben Temperton[38]

**Metadata curation, sequencing and analysis, and visualization**
Sharif Shaaban[59] & Andrew R. Hesketh[28]

**Project administration, samples and logistics, and sequencing and analysis**
Kenneth G. Laing[29], Irene M. Monahan[29] & Judith Heaney[19,20,60]

**Project administration, samples and logistics, and visualization**
Emanuela Pelosi[24], Siona Silviera[24] & Eleri Wilson-Davies[24]

**Samples and logistics, software and analysis tools, and visualization**
Helen Fryer[6]

**Sequencing and analysis, software and analysis tools, and visualization**
Helen Adams[72], Louis du Plessis[63], Rob Johnson[7], William T. Harvey[31,73], Joseph Hughes[31], Richard J. Orton[31], Lewis G. Spurgin[74], Yann Bourgeois[58], Chris Ruis[32], Áine O'Toole[17], Marina Gourtovaia[2] & Theo Sanderson[2]

**Funding acquisition, and leadership and supervision**
Christophe Fraser[6], Jonathan Edgeworth[18], Judith Breuer[19,75], Stephen L. Michell[38] & John A. Todd[76]

**Funding acquisition and project administration**
Michaela John[77] & David Buck[76]

**Leadership and supervision, and metadata curation**
Kavitha Gajee[35] & Gemma L. Kay[52]

**Leadership and supervision, and project administration**
Sharon J. Peacock[4,10] & David Heyburn[14]

**Leadership and supervision, and samples and logistics**
Katie Kitchman[35], Alan McNally[15,78], David T. Pritchard[65], Samir Dervisevic[79], Peter Muir[4], Esther Robinson[4,61], Barry B. Vipond[4], Newara A. Ramadan[80], Christopher Jeanes[81], Danni Weldon[2], Jana Catalan[82] & Neil Jones[82]

**Leadership and supervision, and sequencing and analysis**
Ana da Silva Filipe[31], Chris Williams[14], Marc Fuchs[33], Julia Miskelly[33], Aaron R. Jeffries[38], Karen Oliver[2] & Naomi R. Park[2]

**Metadata curation, and samples and logistics**
Amy Ash[83], Cherian Koshy[83], Magdalena Barrow[84], Sarah L. Buchan[84], Anna Mantzouratou[84], Gemma Clark[85], Christopher W. Holmes[86], Sharon Campbell[87], Thomas Davis[88], Ngee Keong Tan[71], Julianne R. Brown[75], Kathryn A. Harris[75,89], Stephen P. Kidd[55], Paul R. Grant[60], Li Xu-McCrae[61], Alison Cox[68,90], Pinglawathee Madona[68,90], Marcus Pond[68,90], Paul A. Randell[68,90], Karen T. Withell[91], Cheryl Williams[92], Clive Graham[93], Rebecca Denton-Smith[94], Emma Swindells[94], Robyn Turnbull[94], Tim J. Sloan[95], Andrew Bosworth[4,61], Stephanie Hutchings[4], Hannah M. Pymont[4], Anna Casey[96], Liz Ratcliffe[96], Christopher R. Jones[38,97], Bridget A. Knight[38,97], Tanzina Haque[56], Jennifer Hart[56], Dianne Irish-Tavares[56], Eric Witele[56], Craig Mower[57], Louisa K. Watson[57], Jennifer Collins[66], Gary Eltringham[66], Dorian Crudgington[36], Ben Macklin[36], Miren Iturriza-Gomara[62], Anita O. Lucaci[62] & Patrick C. McClure[98]

**Metadata curation, and sequencing and analysis**
Matthew Carlile[40], Nadine Holmes[40], Christopher Moore[40], Nathaniel Storey[75], Stefan Rooke[59],

Gonzalo Yebra[59], Noel Craine[14], Malorie Perry[14], Nabil-Fareed Alikhan[52], Stephen Bridgett[33], Kate F. Cook[12], Christopher Fearn[12], Salman Goudarzi[12], Ronan A. Lyons[99], Thomas Williams[17], Sam T. Haldenby[62], Jillian Durham[2] & Steven Leonard[2]

**Metadata curation, and software and analysis tools**
Robert M. Davies[2]

**Project administration, and samples and logistics**
Rahul Batra[18], Beth Blane[10], Moira J. Spyer[19,20,21], Perminder Smith[8,100], Mehmet Yavus[48,101], Rachel J. Williams[19], Adhyana I. K. Mahanama[24], Buddhini Samaraweera[24], Sophia T. Girgis[32], Samantha E. Hansford[48], Angie Green[76], Charlotte Beaver[2], Katherine L. Bellis[2,32], Matthew J. Dorman[2], Sally Kay[2], Liam Prestwood[2] & Shavanthi Rajatileka[2]

**Project administration, and sequencing and analysis**
Joshua Quick[15]

**Project administration, and software and analysis tools**
Radoslaw Poplawski[15]

**Samples and logistics, and sequencing and analysis**
Nicola Reynolds[102], Andrew Mack[13], Arthur Morriss[13], Thomas Whalley[13], Bindi Patel[18], Iliana Georgana[23], Myra Hosmillo[23], Malte L. Pinckert[23], Joanne Stockton[15], John H. Henderson[44], Amy Hollis[44], William Stanley[44], Wen C. Yew[44], Richard Myers[4], Alicia Thornton[4], Alexander Adams[14], Tara Annett[14], Hibo Asad[14], Alec Birchley[14], Jason Coombes[14], Johnathan M. Evans[14], Laia Fina[14], Bree Gatica-Wilcox[14], Lauren Gilbert[14], Lee Graham[14], Jessica Hey[14], Ember Hilvers[14], Sophie Jones[14], Hannah Jones[14], Sara Kumziene-Summerhayes[14], Caoimhe McKerr[14], Jessica Powell[14], Georgia Pugh[14], Sarah Taylor[14], Alexander J. Trotter[52], Charlotte A. Williams[19], Leanne M. Kermack[32], Benjamin H. Foulkes[48], Marta Gallis[48], Hailey R. Hornsby[48], Stavroula F. Louka[48], Manoj Pohare[48], Paige Wolverson[48], Peijun Zhang[48], George MacIntyre-Cockett[76], Amy Trebes[76], Robin J. Moll[2], Lynne Ferguson[103], Emily J. Goldstein[103], Alasdair Maclean[103] & Rachael Tomb[103]

**Samples and logistics, and software and analysis tools**
Igor Starinskij[31]

**Sequencing and analysis, and software and analysis tools**
Laura Thomson[6], Joel Southgate[13,14], Moritz U. G. Kraemer[63], Jayna Raghwani[63], Alex E. Zarebski[63], Olivia Boyd[7], Lily Geidelberg[7], Chris J. Illingworth[64], Chris Jackson[64], David Pascall[64], Sreenu Vattipally[31], Timothy M. Freeman[48], Sharon N. Hsu[48], Benjamin B. Lindsey[48], Keith James[2], Kevin Lewis[2], Gerry Tonkin-Hill[2] & Jaime M. Tovar-Corona[2]

**Sequencing and analysis, and visualization**
MacGregor Cox[10]

**Software and analysis tools, and visualization**
Khalil Abudahab[2,46], Mirko Menegazzo[46], Ben E. W. Taylor[2,46], Corin A. Yeats[46], Afrida Mukaddas[31], Derek W. Wright[31], Leonardo de Oliveira Martins[52], Rachel Colquhoun[17], Verity Hill[17], Ben Jackson[17], J. T. McCrone[17], Nathan Medd[17], Emily Scher[17] & Jon-Paul Keatley[2]

**Leadership and supervision**
Tanya Curran[39], Sian Morgan[77], Patrick Maxwell[10], Ken Smith[10], Sahar Eldirdiri[88], Anita Kenyon[88], Alison H. Holmes[68,69], James R. Price[68,69], Tim Wyatt[104], Alison E. Mather[52], Timofey Skvortsov[33] & John A. Hartley[19]

**Metadata curation**
Martyn Guest[13], Christine Kitchen[13], Ian Merrick[13], Robert Munn[13], Beatrice Bertolusso[55], Jessica Lynch[55], Gabrielle Vernet[55], Stuart Kirk[60], Elizabeth Wastnedge[50], Rachael Stanley[79], Giles Idle[105], Declan T. Bradley[33,104], Jennifer Poyner[97] & Matilde Mori[106]

**Project administration**
Owen Jones[13], Victoria Wright[40], Ellena Brooks[10], Carol M. Churcher[10], Mireille Fragakis[10], Katerina Galai[4,10], Andrew Jermy[10], Sarah Judges[10], Georgina M. McManus[10], Kim S. Smith[10], Elaine Westwick[10], Stephen W. Attwood[63], Frances Bolt[68,69], Alisha Davies[14], Elen De Lacy[14], Fatima Downing[14], Sue Edwards[14], Lizzie Meadows[52], Sarah Jeremiah[24], Nikki Smith[48] & Luke Foulser[2]

**Samples and logistics**
Themoula Charalampous[16,18], Amita Patel[18], Louise Berry[85], Tim Boswell[85], Vicki M. Fleming[85], Hannah C. Howson-Wells[85], Amelia Joseph[85], Manjinder Khakh[85], Michelle M. Lister[85], Paul W. Bird[86], Karlie Fallon[86], Thomas Helmer[86], Claire L. McMurray[86], Mina Odedra[86], Jessica Shaw[86], Julian W. Tang[86], Nicholas J. Willford[86], Victoria Blakey[87], Veena Raviprakash[87], Nicola Sheriff[87], Lesley-Anne Williams[87], Theresa Feltwell[10], Luke Bedford[107], James S. Cargill[108], Warwick Hughes[108], Jonathan Moore[109], Susanne Stonehouse[109], Laura Atkinson[75], Jack C. D. Lee[75], Divya Shah[75], Adela Alcolea[8,100], Natasha Ohemeng-Kumi[8,100], John Ramble[8,100], Jasveen Sehmi[8,100], Rebecca Williams[55], Wendy Chatterton[60], Monika Pusok[60], William Everson[35], Anibolina Castigador[110], Emily Macnaughton[110], Kate El Bouzidi[111], Temi Lampejo[111], Malur Sudhanva[111], Cassie Breen[112], Graciela Sluga[91], Shazaad S. Y. Ahmad[4,113], Ryan P. George[113], Nicholas W. Machin[4,113], Debbie Binns[65], Victoria James[65], Rachel Blacow[51], Lindsay Coupland[79], Louise Smith[74], Edward Barton[93], Debra Padgett[93], Garren Scott[93], Aidan Cross[114], Mariyam Mirfenderesky[114], Jane Greenaway[94], Kevin Cole[105], Phillip Clarke[95], Nichola Duckworth[95], Sarah Walsh[95], Kelly Bicknell[49], Robert Impey[49], Sarah Wyllie[49], Richard Hopes[4], Chloe Bishop[4],

Vicki Chalker[4], Ian Harrison[4], Laura Gifford[14], Zoltan Molnar[33], Cressida Auckland[97], Cariad Evans[48,101], Kate Johnson[48,101], David G. Partridge[48,101], Mohammad Raza[48,101], Paul Baker[57], Stephen Bonner[57], Sarah Essex[57], Leanne J. Murray[57], Andrew I. Lawton[115], Shirelle Burton-Fanning[66], Brendan A. I. Payne[66], Sheila Waugh[66], Andrea N. Gomes[116], Maimuna Kimuli[116], Darren R. Murray[116], Paula Ashfield[117], Donald Dobie[117], Fiona Ashford[78], Angus Best[78], Liam Crawford[78], Nicola Cumley[78], Megan Mayhew[78], Oliver Megram[78], Jeremy Mirza[78], Emma Moles-Garcia[78], Benita Percival[78], Leah Ensell[19], Helen L. Lowe[19], Laurentiu Maftei[19], Matteo Mondani[19], Nicola J. Chaloner[37], Benjamin J. Cogger[37], Lisa J. Easton[37], Hannah Huckson[37], Jonathan Lewis[37], Sarah Lowdon[37], Cassandra S. Malone[37], Florence Munemo[37], Manasa Mutingwende[37], Roberto Nicodemi[37], Olga Podplomyk[37], Thomas Somassa[37], Andrew Beggs[118], Alex Richter[118], Claire Cormie[32], Joana Dias[32], Sally Forrest[32], Ellen E. Higginson[32], Mailis Maes[32], Jamie Young[32], Rose K. Davidson[53], Kathryn A. Jackson[62], Lance Turtle[62], Alexander J. Keeley[48], Jonathan Ball[98], Timothy Byaruhanga[98], Joseph G. Chappell[98], Jayasree Dey[98], Jack D. Hill[98], Emily J. Park[98], Arezou Fanaie[119], Rachel A. Hilson[119], Geraldine Yaze[119] & Stephanie Lo[2]

**Sequencing and analysis**
Safiah Afifi[77], Robert Beer[77], Joshua Maksimovic[77], Kathryn McCluggage Masters[77], Karla Spellman[77], Catherine Bresner[13], William Fuller[13], Angela Marchbank[13], Trudy Workman[13], Ekaterina Shelest[11,58], Johnny Debebe[40], Fei Sang[40], Marina Escalera Zamudio[63], Sarah Francois[63], Bernardo Gutierrez[63], Tetyana I. Vasylyeva[63], Flavia Flaviani[120], Manon Ragonnet-Cronin[7], Katherine L. Smollett[73], Alice Broos[31], Daniel Mair[31], Jenna Nichols[31], Kyriaki Nomikou[31], Lily Tong[31], Ioulia Tsatsani[31], Sarah O'Brien[121], Steven Rushton[121], Roy Sanderson[121], Jon Perkins[51], Seb Cotton[50], Abbie Gallagher[50], Elias Allara[4,32], Clare Pearson[4,32], David Bibby[4], Gavin Dabrera[4], Nicholas Ellaby[4], Eileen Gallagher[4], Jonathan Hubb[4], Angie Lackenby[4], David Lee[4], Nikos Manesis[4], Tamyo Mbisa[4], Steven Platt[4], Katherine A. Twohig[4], Mari Morgan[14], Alp Aydin[52], David J. Baker[52], Ebenezer Foster-Nyarko[52], Sophie J. Prosolek[52], Steven Rudder[52], Chris Baxter[33], Silvia F. Carvalho[33], Deborah Lavin[33], Arun Mariappan[33], Clara Radulescu[33], Aditi Singh[33], Miao Tang[33], Helen Morcrette[97], Nadua Bayzid[19], Marius Cotic[19], Carlos E. Balcazar[17], Michael D. Gallagher[17], Daniel Maloney[17], Thomas D. Stanton[17], Kathleen A. Williamson[17], Robin Manley[38], Michelle L. Michelsen[38], Christine M. Sambles[38], David J. Studholme[38], Joanna Warwick-Dugdale[38], Richard Eccles[62], Matthew Gemmell[62], Richard Gregory[62], Margaret Hughes[62], Charlotte Nelson[62], Lucille Rainbow[62], Edith E. Vamos[62], Hermione J. Webster[62], Mark Whitehead[62], Claudia Wierzbicki[62], Adrienn Angyal[48], Luke R. Green[48], Max Whiteley[48], Emma Betteridge[2], Iraad F. Bronner[2], Ben W. Farr[2], Scott Goodwin[2], Stefanie V. Lensing[2], Shane A. McCarthy[2,32], Michael A. Quail[2], Diana Rajan[2], Nicholas M. Redshaw[2], Carol Scott[2], Lesley Shirley[2] & Scott A. J. Thurston[2]

**Software and analysis tools**
Will Rowe[15], Amy Gaskin[14], Thanh Le-Viet[52], James Bonfield[2], Jennifer Liddle[2] & Andrew Whitwham[2]

[11]Centre for Enzyme Innovation, University of Portsmouth, Portsmouth, UK. [12]School of Pharmacy & Biomedical Sciences, University of Portsmouth, Portsmouth, UK. [13]Cardiff University, Cardiff, UK. [14]Public Health Wales, Cardiff, UK. [15]Institute of Microbiology and Infection, University of Birmingham, Birmingham, UK. [16]King's College London, London, UK. [17]University of Edinburgh, Edinburgh, UK. [18]Centre for Clinical Infection and Diagnostics Research, Department of Infectious Diseases, Guy's and St Thomas' NHS Foundation Trust, London, UK. [19]University College London Hospitals NHS Foundation Trust, London, UK. [20]Advanced Pathogen Diagnostics Unit, University College London Hospital, London, UK. [21]Great Ormond Street Institute of Child Health, University College London, London, UK. [22]Department of Infectious Diseases and Microbiology, Cambridge University Hospitals NHS Foundation Trust, Cambridge, UK. [23]Division of Virology, Department of Pathology, University of Cambridge, Cambridge, UK. [24]University Hospital Southampton NHS Foundation Trust, Southampton, UK. [25]School of Health Sciences, University of Southampton, Southampton, UK. [26]School of Medicine, University of Southampton, Southampton, UK. [27]Division of Infection and Immunity, University College London, London, UK. [28]University of Brighton, Brighton, UK. [29]Institute for Infection and Immunity, St George's University of London, London, UK. [30]Infection Care Group, St George's University Hospitals NHS Foundation Trust, London, UK. [31]MRC–University of Glasgow Centre for Virus Research, Glasgow, UK. [32]University of Cambridge, Cambridge, UK. [33]Queen's University Belfast, Belfast, UK. [34]Blackpool Teaching Hospitals NHS Foundation Trust, Blackpool, UK. [35]Hull University Teaching Hospitals NHS Trust, Hull, UK. [36]University Hospitals Dorset NHS Foundation Trust, Poole, UK. [37]University Hospitals Sussex NHS Foundation Trust, Worthing, UK. [38]University of Exeter, Exeter, UK. [39]Belfast Health & Social Care Trust, Belfast, UK. [40]Deep Seq, School of Life Sciences, Queens Medical Centre, University of Nottingham, Nottingham, UK. [41]East Kent Hospitals University NHS Foundation Trust, Canterbury, UK. [42]University of Kent, Canterbury, UK. [43]Hub for Biotechnology in the Built Environment, Northumbria University, Newcastle upon Tyne, UK. [44]Northumbria University, Newcastle upon Tyne, UK. [45]NU-OMICS, Northumbria University, Newcastle upon Tyne, UK. [46]Centre for Genomic Pathogen Surveillance, University of Oxford, Oxford, UK. [47]Public Health England, Cambridge, Cambridge, UK. [48]University of Sheffield, Sheffield, UK. [49]Portsmouth Hospitals University NHS Trust, Portsmouth, UK. [50]NHS Lothian, Edinburgh, UK. [51]NHS Greater Glasgow and Clyde, Glasgow, UK. [52]Quadram Institute Bioscience, Norwich, UK. [53]University of East Anglia, Norwich, UK. [54]University of Oxford, Oxford, UK. [55]Hampshire Hospitals NHS Foundation Trust, Basingstoke, UK. [56]Royal Free London NHS Foundation Trust, London, UK. [57]South Tees Hospitals NHS Foundation Trust, Middlesbrough, UK. [58]School of Biological Sciences, University of Portsmouth, Portsmouth, UK. [59]Public Health Scotland, Edinburgh, UK. [60]Health Services Laboratories, London, UK. [61]Heartlands Hospital, Birmingham, UK. [62]University of Liverpool, Liverpool, UK. [63]Department of Zoology, University of Oxford, Oxford, UK. [64]MRC Biostatistics Unit, University of Cambridge, Cambridge, UK. [65]Microbiology Department, Buckinghamshire Healthcare NHS Trust, Aylesbury, UK. [66]The Newcastle upon Tyne Hospitals NHS Foundation Trust, Newcastle upon Tyne, UK. [67]University of St Andrews, St Andrews, UK. [68]Imperial College Healthcare NHS Trust, London, UK. [69]NIHR Health Protection Research Unit in HCAI and AMR, Imperial College London, London, UK. [70]Cambridge University Hospitals NHS Foundation Trust,

Cambridge, UK. [71]Department of Microbiology, South West London Pathology, London, UK. [72]Betsi Cadwaladr University Health Board, Wrexham, UK. [73]Institute of Biodiversity, Animal Health & Comparative Medicine, Glasgow, UK. [74]Norfolk County Council, Norfolk, UK. [75]Great Ormond Street Hospital for Children NHS Foundation Trust, London, UK. [76]Wellcome Centre for Human Genetics, Nuffield Department of Medicine, University of Oxford, Oxford, UK. [77]Cardiff and Vale University Health Board, Cardiff, UK. [78]Turnkey Laboratory, University of Birmingham, Birmingham, UK. [79]Norfolk and Norwich University Hospitals NHS Foundation Trust, Norwich, UK. [80]Royal Brompton and Harefield Hospitals, London, UK. [81]The Queen Elizabeth Hospital King's Lynn NHS Foundation Trust, King's Lynn, UK. [82]Whittington Health NHS Trust, London, UK. [83]Barking, Havering and Redbridge University Hospitals NHS Trust, London, UK. [84]Bournemouth University, Bournemouth, UK. [85]Clinical Microbiology Department, Queens Medical Centre, Nottingham University Hospitals NHS Trust, Nottingham, UK. [86]Clinical Microbiology, University Hospitals of Leicester NHS Trust, Leicester, UK. [87]County Durham and Darlington NHS Foundation Trust, Durham, UK. [88]Department of Microbiology, Kettering General Hospital, Kettering, UK. [89]Barts Health NHS Trust, London, UK. [90]North West London Pathology, London, UK. [91]Maidstone and Tunbridge Wells NHS Trust, Maidstone, UK. [92]Microbiology, Royal Oldham Hospital, Oldham, UK. [93]North Cumbria Integrated Care NHS Foundation Trust, Carlisle, UK. [94]North Tees and Hartlepool NHS Foundation Trust, Stockton on Tees, UK. [95]Path Links, Northern Lincolnshire and Goole NHS Foundation Trust, Grimsby, UK. [96]Queen Elizabeth Hospital, Birmingham, UK. [97]Royal Devon and Exeter NHS Foundation Trust, Exeter, UK. [98]Virology, School of Life Sciences, Queens Medical Centre, University of Nottingham, Nottingham, UK. [99]Swansea University, Swansea, UK. [100]Viapath, Guy's and St Thomas' NHS Foundation Trust, and King's College Hospital NHS Foundation Trust, London, UK. [101]Sheffield Teaching Hospitals NHS Foundation Trust, Sheffield, UK. [102]Cambridge Stem Cell Institute, University of Cambridge, Cambridge, UK. [103]West of Scotland Specialist Virology Centre, NHS Greater Glasgow and Clyde, Glasgow, UK. [104]Public Health Agency, Belfast, UK. [105]Northumbria Healthcare NHS Foundation Trust, Newcastle upon Tyne, UK. [106]University of Southampton, Southampton, UK. [107]East Suffolk and North Essex NHS Foundation Trust, Colchester, UK. [108]East Sussex Healthcare NHS Trust, St Leonards-on-Sea, UK. [109]Gateshead Health NHS Foundation Trust, Gateshead, UK. [110]Isle of Wight NHS Trust, Newport, UK. [111]King's College Hospital NHS Foundation Trust, London, UK. [112]Liverpool Clinical Laboratories, Liverpool, UK. [113]Manchester University NHS Foundation Trust, Manchester, UK. [114]North Middlesex University Hospital NHS Trust, London, UK. [115]Southwest Pathology Services, Taunton, UK. [116]The Royal Marsden NHS Foundation Trust, London, UK. [117]The Royal Wolverhampton NHS Trust, Wolverhampton, UK. [118]University of Birmingham, Birmingham, UK. [119]Watford General Hospital, Watford, UK. [120]Guy's and St Thomas' Biomedical Research Centre, London, UK. [121]Newcastle University, Newcastle, UK.

## Methods

### Pillar 2 SARS-CoV-2 testing data

Publicly available daily SARS-CoV-2 test result data from testing for the wider population outside the National Health Service (Pillar 2 newCasesBySpecimenDate) were downloaded from https://coronavirus.data.gov.uk/ spanning the date range from 1 September 2020 to 30 June 2021 for 315 English LTLAs (downloaded on 20 July 2021). These data are mostly positive PCR tests, with about 4% of results from lateral flow tests without PCR confirmation. In this dataset, the City of London is merged with Hackney, and the Isles of Scilly are merged with Cornwall due to their small number of inhabitants, thereby reducing the number of English LTLAs from 317 to 315. Population data for each LTLA were downloaded from the Office of National Statistics (ONS; https://www.ons.gov.uk/peoplepopulationandcommunity/populationandmigration/populationestimates/datasets/populationestimatesforukenglandandwalesscotlandandnorthernireland).

### SARS-CoV-2 surveillance sequencing

In total, 281,178 tests (September 2020 to June 2021) were collected as part of random surveillance of positive tests of residents of England from four Pillar 2 Lighthouse laboratories. The samples were collected between 1 September 2020 and 26 June 2021. A random selection of samples was taken, after excluding those that were known to be taken during quarantine of recent travellers, and samples from targeted and local surge testing efforts. The available metadata made this selection imperfect, but these samples should be an approximately random selection of infections in England during this time period, and the large sample size makes our subsequent inferences robust.

We amplified RNA extracts from these tests with $C_t$ < 30 using the ARTIC amplicon protocol (https://www.protocols.io/workspaces/coguk/publications). We sequenced 384-sample pools on Illumina NovaSeq, and produced consensus fasta sequences according to the ARTIC nextflow processing pipeline (https://github.com/connor-lab/ncov2019-artic-nf). Lineage assignments were made using Pangolin[5], according to the latest lineage definitions at the time, except for B.1.617, which we reanalysed after the designation of sublineages B.1.617.1, B.1.617.2 and B.1.617.3. Lineage prevalence was computed from 281,178 genome sequences. The genomes were mapped to the same 315 English LTLAs as for the testing data described above. Mapping was performed from outer postcodes to LTLA, which can introduce some misassignment to neighbouring LTLAs. Furthermore, lineages in each LTLA were aggregated to counts per week for a total of 43 weeks, defined beginning on Sunday and ending on Saturday.

Finally, the complete set of 328 SARS-CoV-2 PANGO lineages was collapsed into $l$ = 71 lineages using the underlying phylogenetic tree, such that each resulting lineage constituted at least 100 genomes, unless the lineage has been designated a VOC, variant under investigation (VUI) or variant in monitoring by Public Health England[32].

### Spatiotemporal genomic surveillance model

A hierarchical Bayesian model was used to fit local incidence data in a given day in each local authority and jointly estimate the relative historical prevalence and transmission parameters. In the following, $t$ denotes time and is measured in days. We use the convention that bold lowercase symbols, such as **b**, indicate vectors.

### Motivation

Suppose that $\mathbf{x}'(t) = (\mathbf{b} + r_0(t)) \cdot \mathbf{x}(t)$ describes the ordinary differential equation (ODE) for the viral dynamics for a set of $l$ different lineages. Here $r_0(t)$ is a scalar time-dependent logarithmic growth rate that is thought to reflect lineage-independent transmission determinants, which changes over time in response to behaviour, non-pharmaceutical interventions (NPIs) and immunity. This reflects a scenario in which the lineages differ only in terms of the intensity of transmission, but

not the intergeneration time distribution. The ODE is solved by $\mathbf{x}(t) = e^{\mathbf{c} + \mathbf{b}t + \int_{t_0}^{t} r_0(t)dt} = e^{\mathbf{c} + \mathbf{b}t} v(t)$. The term $v(t)$ contributes the same factor to each lineage and therefore drops from the relative proportions of lineages $\mathbf{p}(t) = \frac{\mathbf{x}(t)}{\sum \mathbf{x}(t)} \propto e^{\mathbf{c} + \mathbf{b}t}$.

In the given model, the lineage prevalence $\mathbf{p}(t)$ follows a multinomial logistic-linear trajectory. Moreover, the total incidence factorizes into $\mu(t) = v(t) \sum e^{\mathbf{c} + \mathbf{b}t}$, which provides a basis to separately estimate the total incidence $\mu(t)$ from Pillar 2 test data and lineage-specific prevalence $\mathbf{p}(t)$ from genomic surveillance data (which are taken from a varying proportion of positive tests). By using the equations above, one can subsequently calculate lineage-specific estimates by multiplying $\mu(t)$ with the respective genomic proportions $\mathbf{p}(t)$.

### Incidence

In the following text, we describe a flexible semi-parametric model of the incidence. Let $\mu(t)$ be the expected daily number of positive Pillar 2 tests and $s$ the population size in each of 315 LTLAs. Denote $\lambda(t) = \log \mu(t) - \log(s)$ the logarithmic daily incidence per capita at time $t$ in each of the 315 LTLAs.

Suppose $f(t)$ is the daily number of new infections caused by the number of people infected at time $t$. As new cases are noticed and tested only after a delay $u$ with distribution $g$, the observed number of cases $f^*(t)$ will be given by the convolution

$$f^*(t) = \int_0^\infty g(u)f(t - u)du = (g * f)(t).$$

The time from infection to test is given by the incubation time plus the largely unknown distribution of the time from symptoms to test, which, in England, was required to take place within 5 d of symptom onset. To account for these factors, the log normal incubation time distribution from ref. [46] is scaled by the equivalent of changing the mean by 2 d. The convolution shifts cases approximately 6 d into the future and also spreads them out according to the width of $g$ (Extended Data Fig. 2a).

To parametrize the short- and longer-term changes of the logarithmic incidence $\lambda(t)$, we use a combination of $h$ weekly and $k - h$ monthly cubic basis splines $\mathbf{f}(t) = (f_1(t), ..., f_k(t))$. The knots of the $h$ weekly splines uniformly tile the observation period except for the last 6 weeks.

Each spline basis function is convolved with the time to test distribution $g$, $\mathbf{f}^*(t) = (f_1^*(t), ..., f_k^*(t))$ as outlined above and used to fit the logarithmic incidence. The derivatives of the original basis $\mathbf{f}'(t)$ are used to calculate the underlying growth rates and $R_t$ values, as shown further below. The convolved spline basis $\mathbf{f}^*(t)$ is used to fit the per capita incidence in each LTLA as (Extended Data Fig. 2b):

$$\lambda(t) = \mathbf{B} \times \mathbf{f}^*(t).$$

This implies that fitting the incidence function for each of the $m$ local authorities is achieved by a suitable choice of coefficients $\mathbf{B} \in \mathbb{R}^{m \times k}$, that is one coefficient for each spline function for each of the LTLAs. The parameters $\mathbf{B}$ have a univariate normal prior distribution each, which reads for LTLA $i$ and spline $j$:

$$\mathbf{B}_{i,j} \sim N(0, \sigma_j).$$

The s.d. of the prior regularizes the amplitude of the splines and is chosen as $\sigma_j = 0.2$ for weekly splines and $\sigma_j = 1$ for monthly splines. This choice was found to reduce the overall variance resulting from the high number of weekly splines, meant to capture rapid changes in growth rates, but which can lead to instabilities particularly at the end of the time series, when not all effects of changes in growth rates are observed yet. The less regularized monthly splines reflect trends on the scale of several weeks and are therefore subject to less noise.

Finally, we introduce a term accounting for periodic differences in weekly testing patterns (there are typically 30% lower specimens taken on weekends; Fig. 1a):

$$\tilde{\boldsymbol{\mu}} = \boldsymbol{\mu}(t) \cdot \delta(t),$$

where the scalar $\delta(t) = \delta(t - i \times 7)\ \forall\ i \in \mathbb{N}$ and prior distribution $\delta(t) \sim \text{LogNormal}(0, 1)$ for $t = 1, \ldots, 6$ and $\delta(0) = 1$.

The total incidence was fitted to the observed number of positive daily tests $\mathbf{X}$ by a negative binomial with a dispersion $\omega = 10$. The overdispersion buffers against non-Poissonian uncorrelated fluctuations in the number of daily tests.

$$\mathbf{X}(t) \sim \text{NB}(\tilde{\boldsymbol{\mu}}(t), \omega).$$

The equation above assumes that all elements of $\mathbf{X}(t)$ are independent, conditional on $\tilde{\boldsymbol{\mu}}(t)$.

## Growth rates and $R_t$ values
A convenient consequence of the spline basis of $\log(\boldsymbol{\mu}) = \boldsymbol{\lambda}$, is that the delay-adjusted daily logarithmic growth rate $\mathbf{r}(t) = \boldsymbol{\lambda}'(t)$ of the local epidemic simplifies to:

$$\mathbf{r}(t) = \mathbf{B} \times \mathbf{f}'(t),$$

where $\mathbf{f}'_j(t)$ represents the first derivative of the $j$th cubic spline basis function.

To express the daily growth rate as an approximate reproductive number $R_t$, one needs to consider the distribution of the intergeneration time, which is assumed to be gamma distributed with mean 6.3 d ($\alpha = 2.29, \beta = 0.36$)[46]. The $R_t$ value can be expressed as a Laplace transform of the intergeneration time distribution[47]. Effectively, this shortens the relative time period because the exponential dynamics put disproportionally more weight on stochastically early transmissions over late ones. For reasons of simplicity and being mindful also of the uncertainties of the intergeneration time distribution, we approximate $R_t$ values by multiplying the logarithmic growth rates with a value of $\bar{\tau}_e = 5.1$ d, which was found to be a reasonable approximation to the convolution required to calculate $R_t$ values (denoted here by the lower case symbol $\boldsymbol{\rho}(t)$ in line with our convention for vector-variate symbols and to avoid confusion with the epidemiological growth rate $r_t$),

$$\log(\boldsymbol{\rho}(t)) \approx \frac{\mathrm{d}\log(\boldsymbol{\mu}(t))}{\mathrm{d}t}\bar{\tau}_e = \mathbf{r}(t)\bar{\tau}_e$$

Thus, the overall growth rate scaled to an effective intergeneration time of 5.1 d can be readily derived from the derivatives of the spline basis and the corresponding coefficients. The values derived from the approach are in very close agreement with those of the method of ref. [48], but shifted according to the typical delay from infection to test (Extended Data Fig. 2b).

## Genomic prevalence
The dynamics of the relative frequency $\mathbf{P}(t)$ of each lineage was modelled using a logistic-linear model in each LTLA, as described above. The logistic prevalence of each lineage in each LTLA is defined as $\mathbf{L}(t) = \text{logit}(\mathbf{P}(t))$ This is modelled using the piecewise linear expression

$$\mathbf{L}(t) = \mathbf{C} + \mathbf{b} \cdot \mathbf{t}_+,$$

where $\mathbf{b}$ may be interpreted as a lineage-specific growth advantage and $\mathbf{C}$ as an offset term of dimension (LTLA × lineages). Time $\mathbf{t}_+$ is measured since introduction $\mathbf{t}_0$ and is defined as

$$\mathbf{t}_+ = t - \mathbf{t}_0 \quad \text{if } t > \mathbf{t}_0 \text{ else} - \infty$$

and accounts for the fact that lineages can be entirely absent prior to a stochastically distributed time period preceding their first observation. This is because, in the absence of such a term, the absence of a lineage prior to the point of observation can only be explained by a higher growth rate compared with the preceding lineages, which may not necessarily be the case. As the exact time of introduction is generally unknown, a stochastic three-week period of $\mathbf{t}_0 \sim \text{Unif}(-14, 0) + \mathbf{t}_0^{\text{obs}}$ prior to the first observation $\mathbf{t}_0^{\text{obs}}$ was chosen.

As the inverse logit transformation projects onto the $l - 1$ dimensional simplex $S_{l-1}$ and therefore loses one degree of freedom, B.1.177 was set as a baseline with

$$\mathbf{L}_{\cdot,0}(t) = 0.$$

The offset parameters $C$ are modelled across LTLAs as independently distributed multivariate normal random variables with a lineage-specific mean $\mathbf{c}$ and covariance $\Sigma = 10 \cdot I_{l-1}$, where $I_{l-1}$ denotes an $(l-1) \times (l-1)$ identity matrix. The lineage-specific parameters growth rate $\mathbf{b}$ and average offset $\mathbf{c}$ are modelled using IID Normal prior distributions

$$\mathbf{b} \sim N(0, 0.2)$$

$$\mathbf{c} \sim N(-10, 5)$$

The time-dependent relative prevalence $\mathbf{P}(t)$ of SARS-CoV2 lineages was fitted to the number of weekly genomes $\mathbf{Y}(t)$ in each LTLA by a Dirichlet-multinomial distribution with expectation $\mathbb{E}[\mathbf{Y}(t)] \approx \mathbf{P}(t) \cdot \mathbf{G}(t)$ where $\mathbf{G}(t)$ are the total number of genomes sequenced from each LTLA in each week. For LTLA $i$, this is defined as:

$$\mathbf{Y}_{i,\cdot}(t) \sim \text{DirMult}(\alpha_0 + \boldsymbol{\alpha}_1 \mathbf{P}_{i,\cdot}(t), \mathbf{G}_i(t)).$$

The scalar parameter $\alpha_0 = 0.01$ can be interpreted as a weak prior with expectation $1/n$, making the model less sensitive to the introduction of single new lineages, which can otherwise exert a very strong effect. Furthermore, the array $\boldsymbol{\alpha}_1 = \frac{\text{cases}}{2}$ increases the variance to account for the fact that, especially at high sequencing coverage (genomes ≈ cases), cases and therefore genomes are likely to be correlated and overdispersed as they may derive from a single transmission event. Other choices such as $\boldsymbol{\alpha}_1 = 1,000$, which make the model converge to a standard multinomial, leave the conclusions qualitatively unchanged. This model aspect is illustrated in Extended Data Fig. 2c.

## Lineage-specific incidence and growth rates
From the two definitions above it follows that the lineage-specific incidence is given by multiplying the total incidence in each LTLA $\boldsymbol{\mu}(t)$ with the corresponding lineage frequency estimate $\mathbf{P}(t)$ for lineage $j$ at each time point

$$\mathbf{M}_{\cdot,j}(t) = \boldsymbol{\mu}(t) \cdot \mathbf{P}_{\cdot,j}(t) \text{ for } j = 0, \ldots, l-1$$

Further corresponding lineage-specific $R_t$ values $\mathbf{R}(t)$ in each LTLA can be calculated from the lineage-agnostic average $R_t$ value $\boldsymbol{\rho}(t)$ and the lineage proportions $\mathbf{P}(t)$ as

$$\log\mathbf{R}(t) = \log\boldsymbol{\rho}(t) + \bar{\tau}_e(\mathbf{b} - \mathbf{P}(t) \times \mathbf{b})$$

By adding the log-transformed growth rate fold changes $\mathbf{b}$ and subtracting the average log-transformed growth rate change $\mathbf{P}(t) \times \mathbf{b}$, it follows that $\mathbf{R}_{i,\cdot}(t) = \mathbf{R}_{i,0}(t)e^{\bar{\tau}_e\mathbf{b}}$, where $\mathbf{R}_{i,0}(t)$ is the $R_t$ value of the reference lineage $j = 0$ (for which $\mathbf{b}_0 = 0$) in LTLA $i$. It follows that all other lineage-specific the $R_t$ values are proportional to this baseline at any given point in time with factor $e^{\bar{\tau}_e\mathbf{b}}$.

## Inference

The model was implemented in numpyro[49,50] and fitted using stochastic variational inference[51]. Guide functions were multivariate normal distributions for each row (corresponding to an LTLA) of $\mathbf{B}$, $\mathbf{C}$ to preserve the correlations across lineages and time as well as for ($\mathbf{b}$, $\mathbf{c}$) to also model correlations between growth rates and typical introduction.

## Phylogeographic analyses

To infer VOC introduction events into the UK and corresponding clade sizes, we investigated VOC genome sequences from GISAID (https://www.gisaid.org/) available from any country. We downloaded multiple sequence alignments of genome sequences with the release dates 17 April 2021 (for the analysis of the lineages A.23.1, B.1.1.318, B.1.351 and B.1.525) and 5 May 2021 (for the analysis of the B.1.617 sublineages). We next extracted a subalignment from each lineage (according to the 1 April 2021 version of PANGOlin for the 17 April 2021 alignment and the 23 April 2021 version of PANGOlin for the 5 May 2021 alignment) and, for each subalignment, we inferred a phylogeny through maximum likelihood using FastTree2 (v.2.1.11)[52] with the default options and GTR substitution model[53].

On each VOC/VUI phylogeny, we inferred the minimum and maximum number of introductions of the considered SARS-CoV-2 lineage into the UK compatible with a parsimonious migration history of the ancestors of the considered samples; we also measured clade sizes for one specific example parsimonious migration history. We counted only introduction events into the UK that resulted in at least one descendant from the set of UK samples that we considered in this work for our hierarchical Bayesian model; similarly, we measured clade sizes by the number of UK samples considered here included in such clades. Multiple occurrences of identical sequences were counted as separate cases, as this helped us to identify rapid SARS-CoV-2 spread.

When using parsimony, we considered only migration histories along a phylogenetic tree that are parsimonious in terms of the number of migration events from and to the UK (in practice, we collapse all of the non-UK locations into a single one). Furthermore, as SARS-CoV-2 phylogenies present substantial numbers of polytomies, that is, phylogenetic nodes where the tree topology cannot be reconstructed due to a lack of mutation events on certain branches, we developed a tailored dynamic programming approach to efficiently integrate over all possible splits of polytomies and over all possible parsimonious migration histories. The idea of this method is somewhat similar to typical Bayesian phylogeographic inference[54] in that it enables us to at least in part integrate over phylogenetic uncertainty and uncertainty in migration history; however, it also represents a very simplified version of these analyses, more so than ref. [16], as it considers most of the phylogenetic tree as fixed, ignores sampling times and uses parsimony instead of a likelihood-based approach. Parsimony is expected to represent a good approximation in the context of SARS-CoV-2, due to the shortness (both in time and substitutions) of the phylogenetic branches considered[55,56]. The main advantage of our approach is that, owing to the dynamic programming implementation, it is more computationally efficient than Bayesian alternatives, as the most computationally demanding step is the inference of the maximum likelihood phylogenetic tree. This enables us to infer plausible ranges for numbers of introduction events for large datasets and to quickly update our analyses as new sequences become available. The other advantage of this approach is that it enables us to easily customize the analysis and to focus on inferred UK introductions that result in at least one UK surveillance sample, while still making use of non-surveillance UK samples to inform the inferred phylogenetic tree and migration history. Note that possible biases due to uneven sequencing rates across the world[55] apply to our approach as well as other popular phylogeographic methods. Our approach works by traversing the maximum likelihood tree starting from the terminal nodes and ending at the root (postorder traversal).

Here, we define a 'UK clade' as a maximal subtree of the total phylogeny for which all terminal nodes are from the UK, all internal nodes are inferred to be from the UK and at least one terminal node is a UK surveillance sample; the size of a UK clade is defined as the number of UK surveillance samples in it. At each node, using values already calculated for all children nodes (possibly more than two children in the case of a multifurcation), we calculate the following quantities: (1) the maximum and minimum number of possible descendant UK clades of the current node, over the space of possible parsimonious migration histories, and conditional on the current node being UK or non-UK; (2) the number of migration events compatible with a parsimonious migration history in the subtree below the current node, and conditional on the current node being UK or non-UK; (3) the size so far of the UK clade the current node is part of, conditional on it being UK; and (4) a sample of UK clade sizes for the subtree below the node. To calculate these quantities, for each internal node, and conditional on each possible node state (UK or non-UK), we consider the possible scenarios of having 0 of 1 migration events between the internal node and its children nodes (migration histories with more than 1 migration event between the node and its children are surely not parsimonious in our analysis and can be ignored).

To confirm the results of our analyses based on parsimony, we also used the new Bayesian phylogenetic approach Thorney BEAST[16] (https://beast.community/thorney_beast) for VOCs for which it was computationally feasible, that is, excluding B.1.351. For each VOC, we used in Thorney BEAST the same topology inferred with FastTree2 as for our parsimony analysis; we also used treetime[57] v.0.8.2 to estimate a timed tree and branch divergences for use in Thorney BEAST. We used a two-state (UK and non-UK) migration model[54] of migration to infer introductions into the UK but again counted, from the posterior sample trees, only UK clades with at least one UK surveillance sample. We used a Skygrid[58] tree coalescent prior with six time intervals. The comparison of parsimony and Bayesian estimates is shown in Extended Data Fig. 8d.

## ONS infection survey analysis

Data from the cross-sectional infection survey were downloaded from https://www.ons.gov.uk/peoplepopulationandcommunity/healthandsocialcare/conditionsanddiseases/bulletins/coronaviruscovid19infectionsurveypilot/30april2021.

Comparison of ONS incidence estimates with hospitalization, case and death rates was conducted by estimating infection trajectories separately from observed cases, hospitalizations and deaths[59,60], convolving them with estimated PCR detection curves[61], and dividing the resulting PCR prevalence estimates by the estimated prevalence from the ONS Community Infection Survey at the midpoints of the two-week intervals over which prevalence was reported in the survey.

## Maps

Maps were plotted using LTLA shapefiles (https://geoportal.statistics.gov.uk/datasets/69dc11c7386943b4ad8893c45648b1e1), sourced from the ONS, which is licensed under the Open Government Licence v.3.0.

## Limitations

A main limitation of the analysis is that the transmission model is deterministic, whereas the spread of variants is a stochastic process. Although the logistic growth assumption is a consistent estimator of the average transmission dynamics, individual outbreaks may deviate from these averages and therefore produce unreliable estimates.

Stochastic growth effects are accounted for only in terms of (uncorrelated) overdispersion and the offset at the time of the introduction. For these reasons, the estimated growth rates may not accurately reflect the viral transmissibility, especially at a low prevalence. It is therefore important to assess whether consistent growth patterns in multiple independent areas are observed. We note that the posterior distribution

of the growth rates of rare variants tends to be biased to the baseline due to the centred prior.

In its current form, the model accounts for only a single introduction event per LTLA. Although this problem is in part alleviated by the high spatial resolution, which spreads introductions across 315 LTLAs, it is important to investigate whether sustained introductions inflate the observed growth rates, as in the case of the Delta variant or other VOCs and VUIs. This can be achieved by a more detailed phylogeographic assessment and through the assessment of monophyletic sublineages.

Furthermore, there is no explicit transmission modelled from one LTLA to another. As each introduction is therefore modelled separately, this makes the model conservative in ascertaining elevated transmission as single observed cases across different LTLAs can be explained by their introduction.

The inferred growth rates also cannot identify a particular mechanism of altered transmission. Biological mechanisms include a higher viral load, longer infectivity or greater susceptibility. Lineages could potentially differ by their intergeneration time, which would lead to nonlinear scaling. Here we did not find convincing evidence in incidence data for such effects, in contrast to previous reports[23]. However, contact-tracing data indicate that the intergeneration time may be shortening for more transmissible lineages such as Delta[33,62]. Cases of the Beta and Gamma VOCs may have been more intensely contact traced and triggered asymptomatic surge testing in some postcode areas. This may have reduced the observed growth rates relative to other lineages.

Lineages, such as Beta, Gamma or Delta also differ in their ability to evade previous immunity. As immunity changes over time, this might lead to a differential growth advantage over time. It is therefore advisable to assess whether a growth advantage is constant over periods in which immunity changes considerably.

A further limitation underlies the nature of lineage definition and assignment. The PANGO lineage definition[5] assigns lineages to geographical clusters, which have by definition expanded, and this can induce a certain survivor bias, often followed by winner's curse. Another issue results from the fact that very recent variants may not be classified as a lineage despite having grown, which can inflate the growth rate of ancestral lineages over sublineages.

As the total incidence is modelled on the basis of the total number of positive PCR tests, it may be influenced by testing capacity; the total number of tests approximately tripled between September 2020 and March 2021. This can potentially lead to a time trend in recorded cases and therefore baseline $R_t$ values if the access to testing changed, for example, by too few tests being available tests during periods of high incidence, or changes to the eligibility to intermittently test with fewer symptoms. Generally, the observed incidence was in good agreement with representative cross-sectional estimates from the ONS[63,64], except for a period of peak incidence from late December 2020 to January 2021 (Extended Data Fig. 1d). Values after 8 March 2021 need to be interpreted with caution as Pillar 2 PCR testing was supplemented by lateral flow devices, which increased the number of daily tests to more than 1.5 million. Positive cases were usually confirmed by PCR and counted only once.

The modelled curves are smoothed over intervals of approximately 7 d using cubic splines, creating the possibility that later time points influence the period of investigation and cause a certain waviness of the $R_t$ value pattern. An alternative parameterization using piecewise linear basis functions per week (that is, constant $R_t$ values per week) leaves the overall conclusions and extracted parameters broadly unchanged.

### Ethical approval

This study was performed as part of surveillance for COVID-19 under the auspices of Section 251 of the National Health Service Act 2006. It therefore did not require individual patient consent or ethical approval. The COG-UK study protocol was approved by the Public Health England Research Ethics Governance Group.

### Reporting summary

Further information on research design is available in the Nature Research Reporting Summary linked to this paper.

### Data availability

PCR test data are publicly available online (https://coronavirus.data. gov.uk/). A filtered, privacy conserving version of the lineage–LTLA– week dataset is publicly available online (https://covid19.sanger.ac.uk/ downloads) and enables strong reproduction of our results, despite a small number of cells having been suppressed to avoid disclosure. Full SARS-CoV-2 genome data and geolocations can be obtained under controlled access from https://www.cogconsortium.uk/data/. Application for full data access requires a description of the planned analysis and can be initiated at coguk_DataAccess@medschl.cam.ac.uk. The data and a version of the analysis with fewer lineages can be interactively explored at https://covid19.sanger.ac.uk. Source data are provided with this paper.

### Code availability

The genomic surveillance model is implemented in Python and available at GitHub (https://github.com/gerstung-lab/genomicsurveillance) and as a PyPI package (genomicsurveillance). Specific code for the analyses of this study can be found as individual Google colab notebooks in the same repository. These were run using Python v.3.7.1 (packages: matplotlib (v.3.4.1), numpy (v.1.20.2), pandas (v.1.2.3), scikit-learn (v.0.19.1), scipy (v.1.6.2), seaborn (v.0.11.1), jax (v.0.2.8), genomicsurveillance (v.0.4.0), numpyro (v.0.4.0)). The phylogeographic analyses were performed using Thorney Beast (v.0.1.1) and https://github.com/ NicolaDM/phylogeographySARS-CoV-2. Code for the ONS infection survey analysis is available at GitHub (https://github.com/jhellewell14/ ons_severity_estimates).

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

**Acknowledgements** We thank E. Allara (Cambridge) and G. Whitton (Sanger) for providing outer postcodes to LTLA mappings; R. Beale for comments and J. McCrone for setting up Thorney Beast analysis; all of the contributors who submitted genome sequences to GISAID (acknowledgement tables for individual sequences are provided at GitHub; https://github.com/NicolaDM/phylogeographySARS-CoV-2); and our colleagues at EMBL-EBI, the Wellcome Sanger Institute and COG-UK for discussions and comments on this manuscript. COG-UK is supported by funding from the Medical Research Council (MRC), part of UK Research & Innovation (UKRI), the National Institute of Health Research (NIHR) and Genome Research Limited, operating as the Wellcome Sanger Institute. Additional sequence generation was funded by the Department of Health and Social Care. H.S.V., J.P.G. and M.G. are supported by a grant from the Department of Health and Social Care. A.W.J., E.B. and M.G. are beneficiaries from grant NNF17OC0027594 from the Novo Nordisk Foundation. E.V. is supported by Wellcome Trust grant 220885/Z/20/Z. T.S. is supported by grant 210918/Z/18/Z, and J.H. and S.F. by grant 210758/Z/18/Z from the Wellcome Trust. H.S.V., N.D.M., A.W.J., N.G., E.B. and M.G. are supported by EMBL.

**Author contributions** H.S.V. and M.G. developed the analysis code, which H.S.V. implemented with input from A.W.J.; H.S.V. created most of the figures. M.S. analysed, annotated and aggregated viral genome data. N.D.M. conducted phylogeographic analyses supervised by N.G.; T.S., R.G., M.S. and H.S.V. developed the interactive spatiotemporal viewer. T.N., F.S., I.H., R.A., C.A., S.G., D.J., I.J., C.S., J.S., T.S. and M.S. analysed genomic surveillance data under the supervision of D.K., M.C., I.M. and J.C.B.; J.H. and S.F. analysed ONS data and helped with epidemiological modelling and data interpretation. E.V. analysed growth rates and helped with data interpretation. E.B. and J.P.G. supervised H.S.V. and helped with data interpretation. J.C.B. and M.G. supervised the analysis with advice from I.M.; M.G., H.S.V., M.S., N.D.M., T.S., I.M. and J.C.B. wrote the manuscript with input from all of the co-authors.

**Funding** Open access funding provided by Deutsches Krebsforschungszentrum (DKFZ).

**Competing interests** E.B. is a paid consultant of Oxford Nanopore.

**Additional information**
**Correspondence and requests for materials** should be addressed to Jeffrey C. Barrett or Moritz Gerstung.

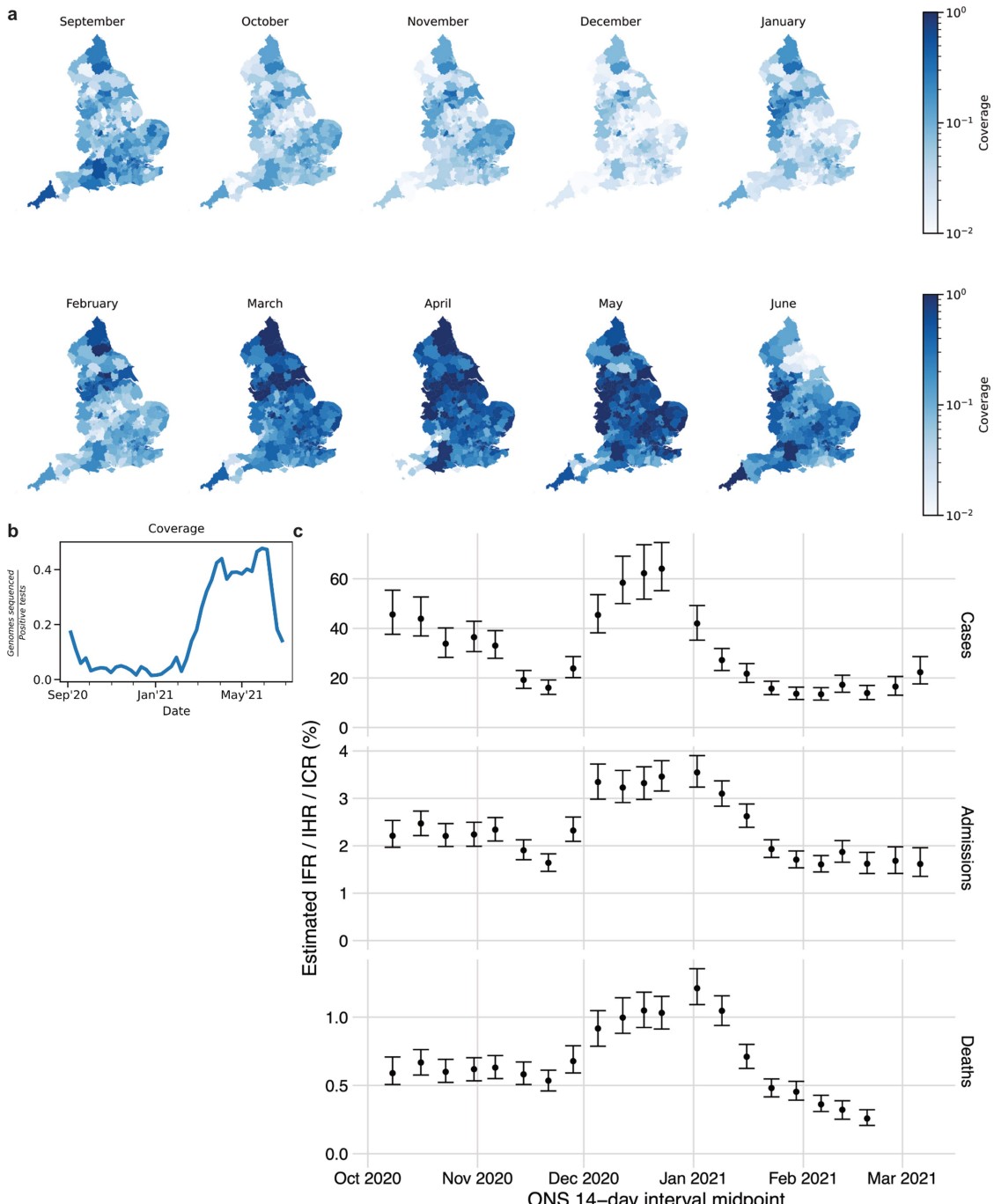

**Extended Data Fig. 1 | SARS-CoV-2 surveillance sequencing in England between September 2020 and June 2021. a**. Local monthly coverage across 315 LTLAs. **b**. Weekly coverage of genomic surveillance sequencing. **c**. Hospitalization, case and infection fatality rates relative to ONS prevalence. Dots denote mean estimates and error bars 95% CIs.

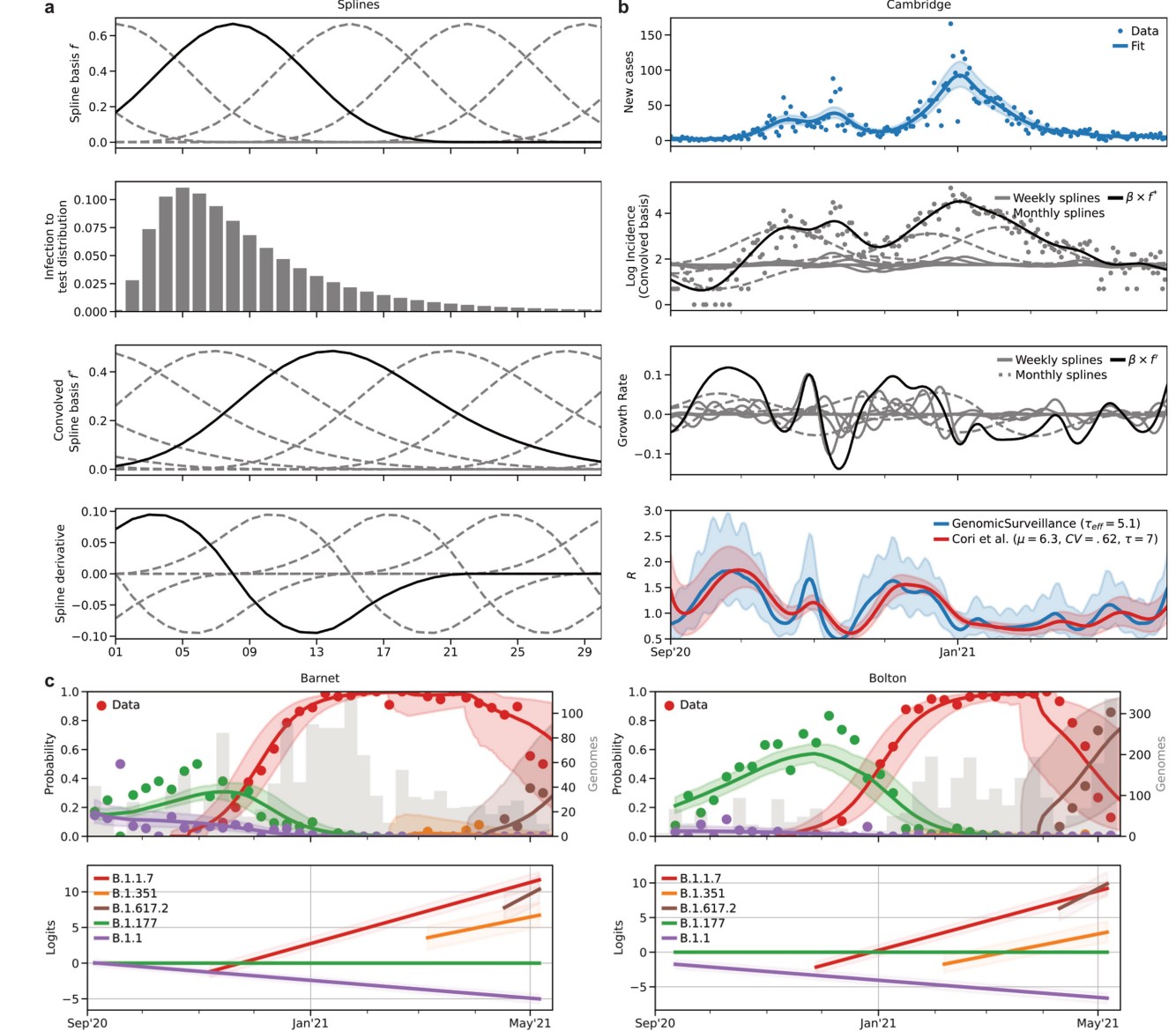

**Extended Data Fig. 2 | Genomic surveillance model of total incidence and lineage-specific frequencies. a**. Cubic basis splines (top row) are convolved with the infection to test distribution (row 2 and 3) and used to fit the log incidence in a LTLA and its corresponding derivatives (growth rates; bottom row). **b**. Example incidence (top row), logarithmic incidence with individual convolved basis functions (dashed lines, row 2), growth rate with individual spline basis derivatives (dashed lines, row 3) and resulting (case) reproduction numbers (growth rate per 5.1d) from our approach (GenomicSurveillance) and estimates by EpiEstim[48], shifted by 10d to approximate a case reproduction number. **c**. The relative frequencies of 62 different lineages are modelled using piecewise multinomial logistic regression. The linear logits are modelled to jump stochastically within 21d prior to first observation to account for the effects of new introductions. Shown are the logits of 5 selected lineages in two different LTLAs.

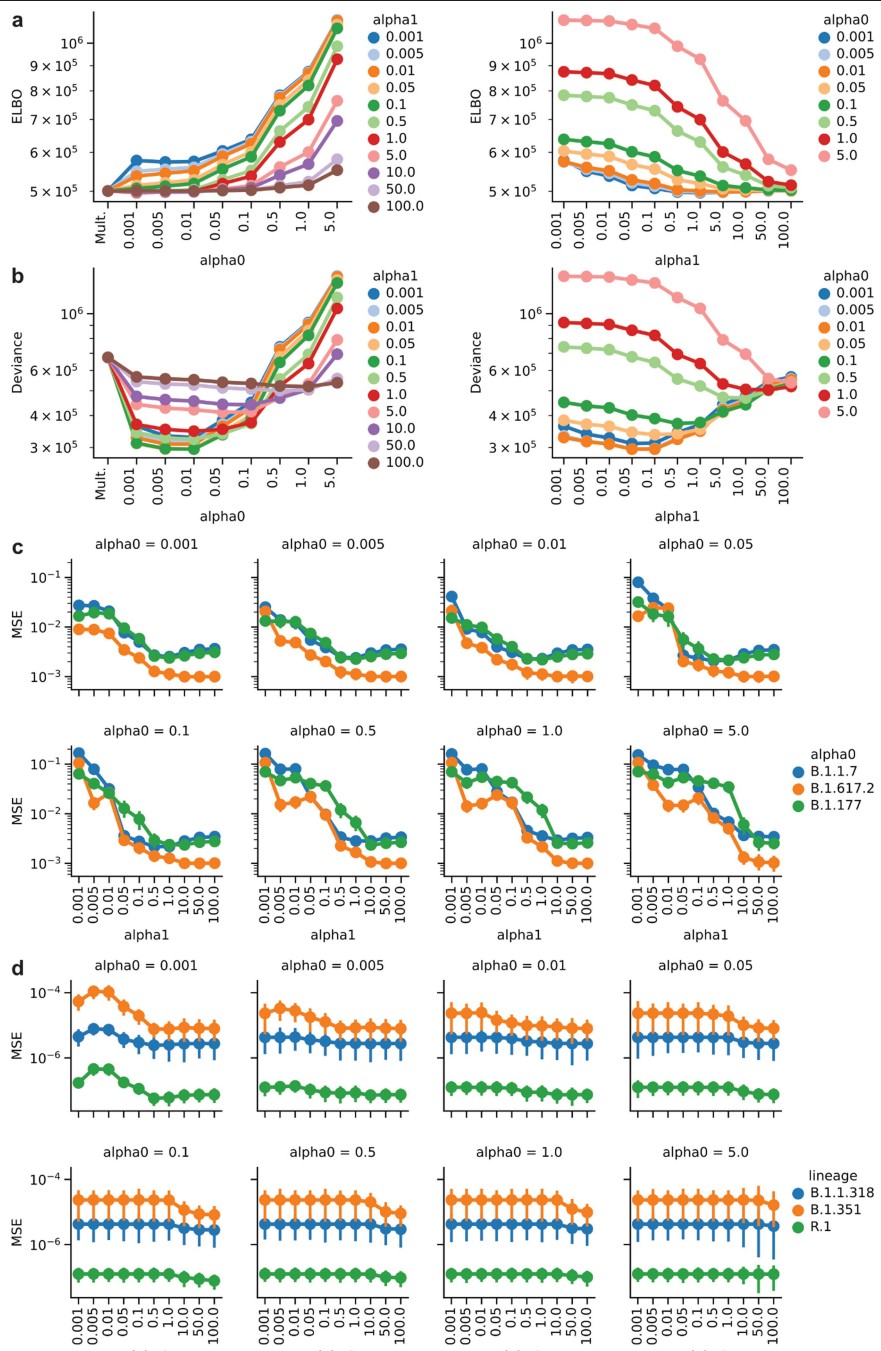

**Extended Data Fig. 3 | Genomic surveillance model selection. a**. Model loss in terms of the ELBO objective function and the model hyperparameters alpha0 and alpha1 (see Methods). **b**. Model deviance (calculated as −2 x log pointwise predictive density) with respect to the model hyperparameters $\alpha_0$ and $\alpha_1$ (see Methods). **c**. Mean squared error (MSE) of modelled weekly proportions of highly prevalent lineages with respect to the model parameters $\alpha_0$ and $\alpha_1$ (see Methods). **d**. Same as in **c**, but for lineages exhibiting low frequencies (VOCs).

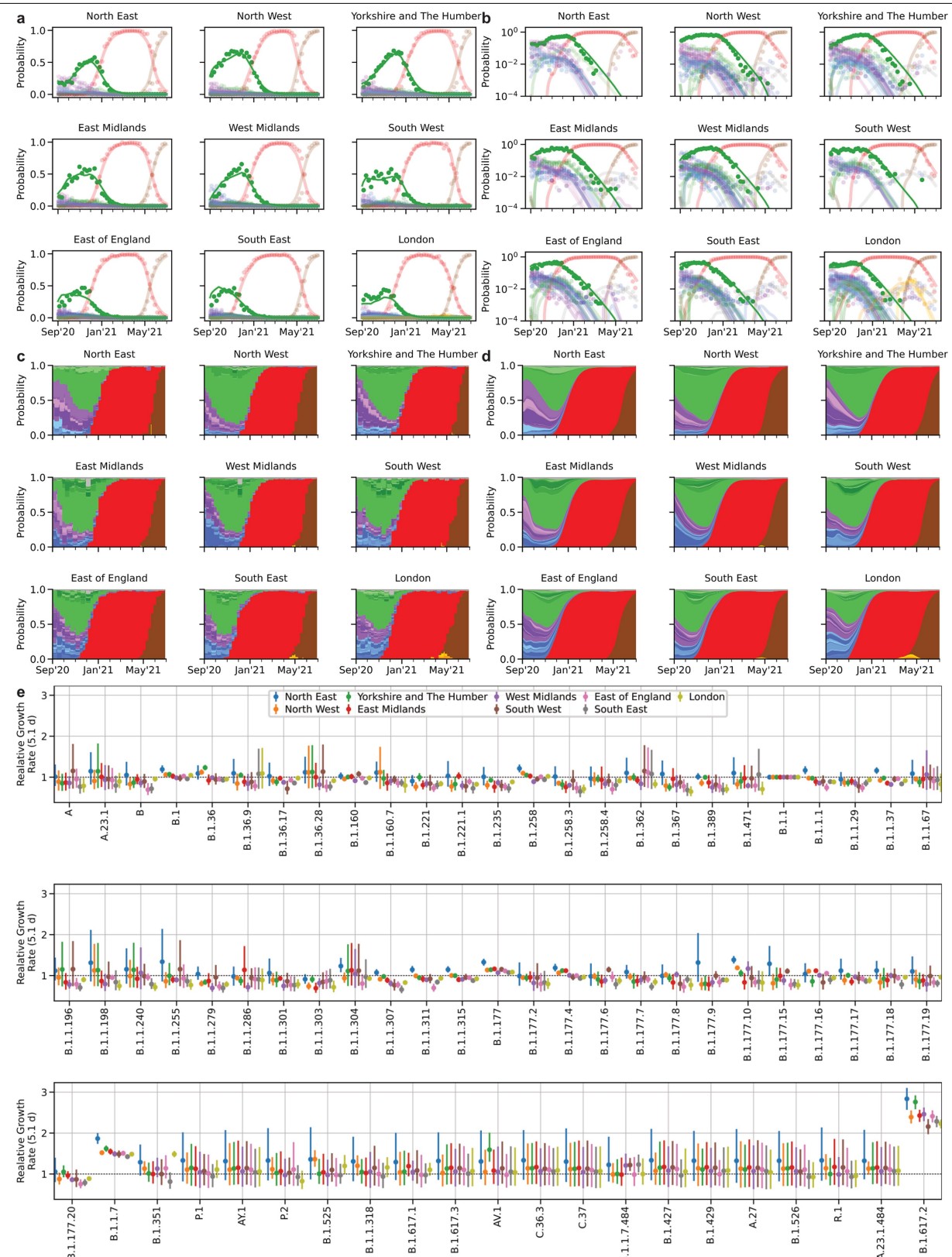

**Extended Data Fig. 4 | Spatiotemporal model of 71 SARS-CoV-2 lineages in 315 English LTLAs between September 2020 and June 2021. a**. Regional lineage specific relative frequency of lineages contributing more than 50 genomes during the time period shown. Dots denote observed data, lines the fits aggregated to each region. **b**. Same as **a**, but on a log scale. **c**. Same data as in **a**, shown as stacked bar charts. Colours resemble major lineages as indicated and shadings thereof indicate sublineages. **d**. Same fits as in **a**, shown as stacked segments. **e**. Average growth rates for 71 SARS-Cov2 lineages estimated in different regions in England. Dots denote median estimates and error bars 95% CIs.

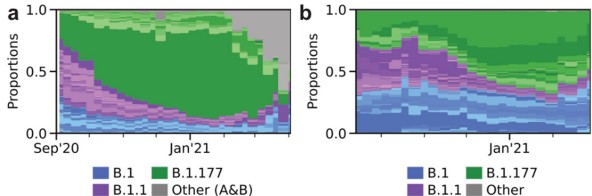

**Extended Data Fig. 5 | Relative growth of B.1.177. a.** Lineage-specific relative frequency data in England, excluding B.1.1.7 and other VOCs/VUIs (Category Other includes: A, A.18, A.20, A.23, A.25, A.27, A.28, B, B.29, B.40, None). Colours resemble major lineages as indicated and shadings thereof indicate sublineages. **b.** Lineage-specific relative frequency data in Denmark, excluding B.1.1.7 and other VOCs/VUIs. Colours resemble major lineages as indicated and shadings thereof indicate sublineages.

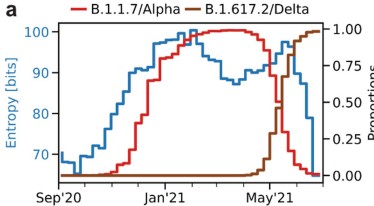

**Extended Data Fig. 6 | Genomic diversity of the SARS-CoV-2 epidemic.**
Shown is the entropy (blue), total number of observed Pango lineages
(grey, divided by 4), as well as the proportion of B.1.1.7 (orange, right axis).
The sweep of B.1.1.7 causes an intermittent decline of genomic diversity as
measured by the entropy.

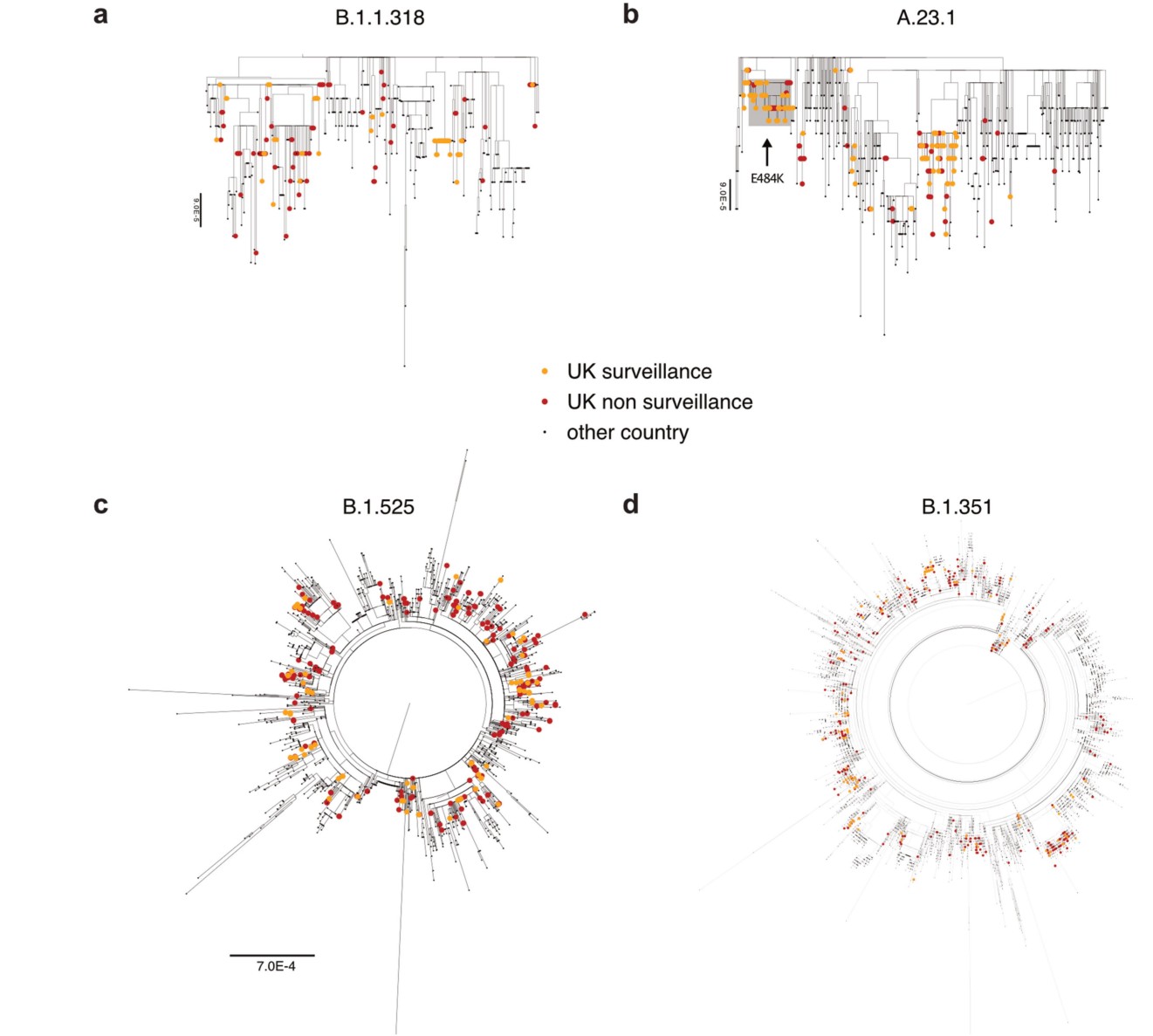

**a** B.1.1.318

**b** A.23.1

E484K

- ● UK surveillance
- ● UK non surveillance
- · other country

**c** B.1.525

7.0E-4

**d** B.1.351

**Extended Data Fig. 7 | Global phylogenetic trees of selected VOCs/VUIs.** English surveillance and other (targeted and quarantine) samples are highlighted respectively orange and red.

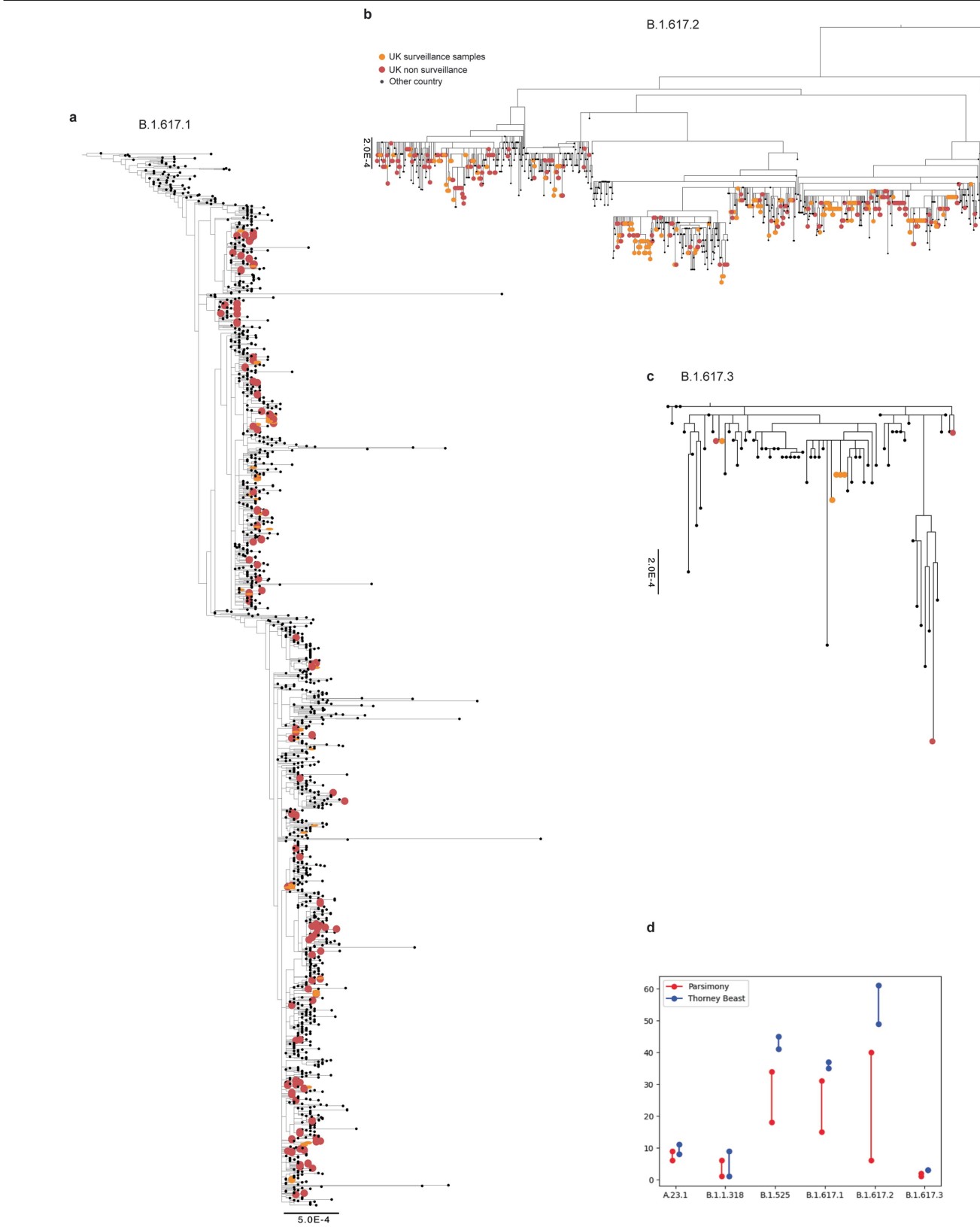

**Extended Data Fig. 8 | Global phylogenetic trees of B.1.617 sublineages.**
**a**, **b and c**. English surveillance and other (targeted and quarantine) samples are highlighted respectively orange and red. The trees of B.1.617.1 and B.1.617.2 are rooted. **d**. Number of UK introductions inferred by parsimony (minimum and maximum numbers) and by Thorney BEAST (95% posterior CI) for each VOC.

# nature research

# Reporting Summary

Nature Research wishes to improve the reproducibility of the work that we publish. This form provides structure for consistency and transparency in reporting. For further information on Nature Research policies, see our Editorial Policies and the Editorial Policy Checklist.

## Statistics

For all statistical analyses, confirm that the following items are present in the figure legend, table legend, main text, or Methods section.

| n/a | Confirmed | |
|---|---|---|
| ☐ | ☒ | The exact sample size (*n*) for each experimental group/condition, given as a discrete number and unit of measurement |
| ☒ | ☐ | A statement on whether measurements were taken from distinct samples or whether the same sample was measured repeatedly |
| ☒ | ☐ | The statistical test(s) used AND whether they are one- or two-sided *Only common tests should be described solely by name; describe more complex techniques in the Methods section.* |
| ☒ | ☐ | A description of all covariates tested |
| ☐ | ☒ | A description of any assumptions or corrections, such as tests of normality and adjustment for multiple comparisons |
| ☐ | ☒ | A full description of the statistical parameters including central tendency (e.g. means) or other basic estimates (e.g. regression coefficient) AND variation (e.g. standard deviation) or associated estimates of uncertainty (e.g. confidence intervals) |
| ☒ | ☐ | For null hypothesis testing, the test statistic (e.g. *F*, *t*, *r*) with confidence intervals, effect sizes, degrees of freedom and *P* value noted *Give P values as exact values whenever suitable.* |
| ☐ | ☒ | For Bayesian analysis, information on the choice of priors and Markov chain Monte Carlo settings |
| ☐ | ☒ | For hierarchical and complex designs, identification of the appropriate level for tests and full reporting of outcomes |
| ☒ | ☐ | Estimates of effect sizes (e.g. Cohen's *d*, Pearson's *r*), indicating how they were calculated |

*Our web collection on statistics for biologists contains articles on many of the points above.*

## Software and code

Policy information about availability of computer code

| Data collection | Consensus Fasta sequences were created using the ARTIC nextflow processing pipeline and SARS-CoV-2 lineage assignments using the Pangolin software (01-04-2021 and 23-04-2021 version) and FastTree2 (2.1.11). |
|---|---|
| Data analysis | Code for spatio-temporal modeling of viral lineages is available at https://github.com/gerstung-lab/genomicsurveillance and as a PyPI package (genomicsurveillance). Analyses were performed in Python 3.7.1 (Packages: matplotlib (3.4.1), numpy (1.20.2), pandas (1.2.3), scikit-learn (0.19.1), scipy (1.6.2), seaborn (0.11.1), jax (0.2.8), genomicsurveillance (0.4.0), numpyro (0.4.0)). The phylogeographic analyses were performed using Thorney Beast (0.1.1) and https://github.com/NicolaDM/phylogeographySARS-CoV-2. Code for ONS infection survey analysis is available at https://github.com/jhellewell14/ons_severity_estimates. |

For manuscripts utilizing custom algorithms or software that are central to the research but not yet described in published literature, software must be made available to editors and reviewers. We strongly encourage code deposition in a community repository (e.g. GitHub). See the Nature Research guidelines for submitting code & software for further information.

## Data

Policy information about availability of data

All manuscripts must include a data availability statement. This statement should provide the following information, where applicable:

- Accession codes, unique identifiers, or web links for publicly available datasets
- A list of figures that have associated raw data
- A description of any restrictions on data availability

PCR test data are publicly available at https://coronavirus.data.gov.uk/.
SARS-CoV-2 genome data and geolocations can be obtained under controlled access from https://www.cogconsortium.uk/data/.

# Field-specific reporting

Please select the one below that is the best fit for your research. If you are not sure, read the appropriate sections before making your selection.

☒ Life sciences      ☐ Behavioural & social sciences      ☐ Ecological, evolutionary & environmental sciences

For a reference copy of the document with all sections, see nature.com/documents/nr-reporting-summary-flat.pdf

# Life sciences study design

All studies must disclose on these points even when the disclosure is negative.

| | |
|---|---|
| Sample size | The study is based on data from 281,178 viral genomes and 3,894,234 positive PCR tests collected in England during the time period from September 1, 2020 to June 30, 2021. This is an observational study based on an existing data set compiled by COG-UK, therefore no sample size calculation is applicable. |
| Data exclusions | No data was excluded in the analysis. |
| Replication | This is an observational study based on an existing data set compiled by COG-UK, therefore no replication is applicable. |
| Randomization | This is an observational study based on an existing data set compiled by COG-UK, therefore no randomization is applicable. |
| Blinding | This is an observational study based on an existing data set compiled by COG-UK, therefore no blinding is applicable. |

# Reporting for specific materials, systems and methods

We require information from authors about some types of materials, experimental systems and methods used in many studies. Here, indicate whether each material, system or method listed is relevant to your study. If you are not sure if a list item applies to your research, read the appropriate section before selecting a response.

### Materials & experimental systems

| n/a | Involved in the study |
|---|---|
| ☒ ☐ | Antibodies |
| ☒ ☐ | Eukaryotic cell lines |
| ☒ ☐ | Palaeontology and archaeology |
| ☒ ☐ | Animals and other organisms |
| ☒ ☐ | Human research participants |
| ☒ ☐ | Clinical data |
| ☒ ☐ | Dual use research of concern |

### Methods

| n/a | Involved in the study |
|---|---|
| ☒ ☐ | ChIP-seq |
| ☒ ☐ | Flow cytometry |
| ☒ ☐ | MRI-based neuroimaging |

