## [Peer Review File · Nature]

Manuscript Title: Genomic reconstruction of the SARS-CoV-2 epidemic in England

Reviewer Comments & Author Rebuttals

Reviewer Reports on the Initial Version:

Referee #1 (Remarks to the Author):

This is a very timely and interesting manuscript. In general, the analysis presented is of very good quality. However, the manuscript gives the impression of three separate analyses of the dataset and that the authors rushed to put the final manuscript.

In order to improve the manuscript, I suggest that the authors select the same date for the final analysis of the dataset and use this date across all figures, as at the moment it is quite confusing. For example, the first part of the manuscript focused on the expansion of B.1.1.7 and ends in March 2021. The second part mentions the replacement of B.1.1.7 with VOCs with 484K, which with hindsight, may not really have happened as the Delta (B.1.617.2) has taken over. The figures are also difficult to compare as they use different data and dates, for example, figures 1, 2 and 3 dates go until March 2021, but Figure 4 dates end up in May 2021 (Figure 4a) and April 2021 (Figure 4b). Figure 5 is even more confusing as different panels (A-E) present data until March, April, and May 2021.

The authors also need to decide the LTLAs that they are discussing in the manuscript. At present, 50 selected LTLAs are presented in Figure 1 but only six LTLAs in Figure 2 and four in LTLAs Figure 3. In addition, given that the LTLAs selected are not the same between Figure 2 and 3, it gives the impression to the reader that the authors may be 'cherry-picking' the LTLAs that fit their hypotheses. I personally do not believe that this is the case but maybe good to select a similar number of LTLAs in Figures 2 and 3 and to try to select the same LTLAs or provide a rationale for the selection of the LTLAs for the figures.

The discussion is also very confusing with all of the different endpoints (March, April, May 2021). The discussion section also lacks a paragraph on the limitation of the analyses of the data. In addition, I was surprised to see the conclusion of the manuscript, which is not in line with the results presented or discussed:

“The global presence of SARS-CoV-2 in human populations, the existence of human-to-animal and animal-to-animal transmission means it is unlikely that global eradication is possible at least in the short to mid term. Therefore a global perspective on controlling and surveilling the virus is essential.”

In summary, my main suggestion is that the author re-write the manuscript with a clear message of how VOCs are replacing each other during the pandemic in England. Many of the analyses will need to be re-done with a unique end date, either on the spring of 2021 or the summer of 2021. Once all the figures relate to each other, we may end up with a superb manuscript that describes how the more transmissible VOCs replaced it other in England and how restrictions affect the epidemic over time.

Referee #2 (Remarks to the Author):

The manuscript by Vöhringer and colleagues documents the genomic epidemiology of SARS-CoV-2 in England during a critical time period of the COVID-19 pandemic. The authors elegantly demonstrate how lineage information can be integrated with incidence data in a spatio-temporal surveillance model, delivering insights with a remarkable resolution, including estimates of lineage-specific growth rates. Although the general picture of lineage turnover and its impact on transmission dynamics was already evident thanks to near real-time surveillance efforts, this study also takes advantage of the dense sampling to uncover more detailed dynamics ‘under the surface’ such as the low-level persistence or slow rise of variants with the E484K spike mutation from December to April. While various studies have already highlighted the tremendous efforts by the COG-UK and its exemplary role in SARS-COV-2 surveillance, I believe the current study may set yet another standard for modern large-scale genomic epidemiology. The authors should also be commended for the short turnaround time in putting together the current study.

At the core of this study stands a state-of-the-art spatio-temporal surveillance model that integrates incidence and genomic data. The Authors adequately outline the limitations of the model but ask the reader to visually assess model fit (in Figure 2 b & c, and Supplementary Figure 3). While I do not wish to question a good fit, in particular at the level of larger regions, a more formal assessment would be desirable. I am wondering for example whether a cross-validation approach could be applicable in this setting.

Given the fast-evolving epidemic dynamics and the communication of findings from near real-time surveillance, it is likely that the recent rapid rise of B.1.617.2 may receive most of the attention. The Authors present a growth estimate for B.1.617.2 of about 37% in excess of that of B.1.1.7, but

remain nuanced in their interpretation by pointing at different factors that could be responsible for this. Data that has accumulated more recently may help to shed further light on this. I would not want to insist to update this analysis as manuscripts will always lag behind the latest available data. However, the authors may have anticipated this and closely followed up the situation, in which case an update may be not be too cumbersome.

Minor text errors:

Nevertheless B.1.1.7 nevertheless

Figure 5, caption: ...in two selected LTLAs -> in four selected LTLAs?

Methods, Limitations: especially a low prevalence -> especially at low prevalence?

Referee #3 (Remarks to the Author):

A. Summary of key results

The authors do a very nice job of providing a summary of their key results via four "chapters" in their Discussion. This includes that: 1) the intro of B.1.177 became dominant with a peak in October 2020; 2) B.1.1.7 became dominant via a selective sweep of other lineages; 3) A December 2020 was needed to slow B.1.1.7 which also contracted other lineages that were eliminated in Spring 2021; 4) the E484K mutation in the Spike protein were introduced repeatedly from December to April 2021.

B. Originality and significance

The work is original and significant and leverages the strength of the UK Genomic Consortium and includes a sequencing rate of over 5% of cases.

C. Data & methodology

The work includes mathematical modeling and phylogenetic analysis. It was well done but some concerns:

1) It was surprising that no vaccination data was considered even in the later months of the models and in May 2021.

2) For phylogenetics, were duplicate weekly sequences by LTLA removed?

3) More description on the parsimony approach for introductions. Why was Bayesian phylogenetics with ancestral state reconstruction not considered? There are new approaches for analyzing larger datasets (https://beast.community/thorney_beast).

4) What non-UK countries were included? That should be clearer in the main manuscript.

D. Appropriate use of statistics and treatment of uncertainties

No concerns

E. Conclusions: robustness, validity, reliability

Discussion section needs improvement.

1) No consideration of other literature given the implications of the study findings.

2) No Limitations paragraph

3) Now that we have these results, what do these results suggest for UK to best protect themselves?

F. Suggested improvements

In addition to the concerns listed above.

1) Suggest adding Greek letters in addition to Pangolin lineages to be consistent with new standards.

2) Better explanation of the scientific premise (a priori hypothesis) before starting the study. What were the key questions that needed to be answered.

3) Explain the choice for the study period.

4) Figure 1C. Suggest a color-blind check of the colors used for the heat map.

5) Figure 2B-C. Relative frequencies really small and hard to visualize. Removing panel d might help a little or making one of them separate.

6) "Furthermore, at low frequency growth can be stochastic..." remove "at"

G. References

Appropriate

H. Clarity and context

Main concern was with Discussion. Already addressed.

Author Rebuttals to Initial Comments:

Rebuttal letter

We thank all three reviewers for their positive and constructive comments, which helped us further improve the manuscript.

As there were a series of overlapping comments, we first summarise the main changes. These are

1. We **extended the data set until June 26, 2021** to cover the entire sweep of the Delta variant in England. This data set, including approximately 100,000 additional genomes, is consistently used throughout the entire study. In line with this, **Figures 1-5** and **Extended Data Figures 1, 3, 4, 6** have been updated.
2. The availability of further data points has raised the **growth rate advantage of Delta over Alpha to 1.59**. We discuss Delta's growth advantage derived from increased transmissibility (higher SAR, spread in young and unvaccinated groups) and immune escape (reduced neutralisation and vaccine effectiveness, elevated reinfection) based on a number of emerging reports.
3. We have **rewritten the manuscript, especially the Discussion**, to clearly highlight the key insight how the two selective sweeps by the Alpha and Delta variants outgrew restrictions containing previous lineages. The main text has been shortened to 3,000 words for a focused exposition. We have also shortened the title to "*Genomic reconstruction of the SARS-CoV-2 epidemic in England*", in line with the journal's style requirements.
4. We use **Greek letters** to label variants designated by the WHO, as this has become the new standard

Please find detailed answers to individual comments below. These include a quantitative assessment of the quality of fit (new **Extended Data Figure 3; 4c**), comparison to a Bayesian phylogeographic analysis (new **Extended Data Figure 8d**) and also a more systematic selection of LTLAs shown in the main Figures and a new **Supplementary Note 1 & 2** showing the fits of all LTLAs.

As part of the reanalysis, we have also changed the parameterisation of the spline fit to a combination of weekly and monthly splines, which provided a more stable fit and avoid overly large R_t values in some LTLAs with low incidence; the model also adjusts for differences in PCR tests by weekday. These minor amendments leave the overall conclusions unchanged.

Figure 1. SARS-CoV-2 surveillance sequencing in England between September 2020 and June 2021. **a.** Positive Pillar 2 SARS-CoV-2 tests in England. **b.** Relative frequency of 328 different PANGO lineages, representing approximately 7.2% of tests shown in **a.** **c.** Positive tests (top row) and frequency of 4 major lineages across 315 English lower tier local authorities. **d.** Absolute frequency of sequenced genomes mapped to 71 PANGO lineages. Blue areas in the pie charts are proportional to the fraction of LTLAs where a given lineage was observed.

Figure 2. Spatiotemporal model of 71 SARS-CoV-2 lineages in 315 English LTLAs between September 2020 and June 2021. a. Average growth rates for 71 lineages. **b.** Lineage specific relative frequency for 35 selected LTLAs, arranged by longitude and latitude to geographically cover England. **c.** Fitted lineage-specific relative frequency for the same LTLAs as **b.** **d.** Fitted lineage-specific incidence for the same LTLAs as in **b.**

Figure 3. Growth of B.1.1.7/Alpha and other lineages in relation to lockdown restrictions between November 2020 and March 2021. **a.** Maps and dates of national and regional restrictions in England. Second national lockdown: closed hospitality businesses, contacts ≤ 2 outdoors only, open schools, reasonable excuse needed for leaving home⁶⁴. Tier 1: private indoor gatherings ≤ 6 persons. Tier 2: as tier 1, restricted hospitality services, gatherings ≤ 6 in public outdoor places. Tier 3: as tier 2, most hospitality businesses closed. Tier 4: as tier 3, single outdoor contact. Third national lockdown: Closed schools with the exception of key workers. **b.** Local lineage-specific R_t values for Alpha and the average R_t value (growth per 5.1d) of all other lineages in the same periods. **c.** Boxplots of R_t values shown in **b**, boxes show quartiles, whiskers extend to 1.5x the inter quartile range. **d.** Total and lineage-specific incidence (top) and R_t values (bottom) for 6 selected LTLAs during the period of restrictions. **e.** Crude lineage-specific fold changes (odds ratios) for Alpha and other lineages across the second (orange) and third national lockdown (red).

Figure 4. Elimination of SARS-CoV-2 lineages during spring 2021. **a** modelled lineage-specific incidence in England. Colors resemble major lineages as indicated and shadings thereof indicate sublineages. **b**. Observed number of PANGO lineages per week.

Figure 5. Dynamics of VOC and VUIs between January and June 2021. **a.** Observed relative frequency of other lineages (light grey), Alpha/B.1.1.7 (dark grey), VOC/VUIs (orange), and Delta/B.1.617.2 (brown). **b.** Observed and modelled relative frequency of VOC/VUIs in England. **c.** Total and relative lineage-specific incidence in four selected LTLAs. **d.** Estimated UK VOC/VUI clade numbers (numbers in square parentheses represent minimum and maximum numbers) and sizes. **e.** Crude growth rates (odds ratios) of Delta and Alpha between April and June 2021, as in **Fig. 3e**. **f.** , Boxplots of lineage-specific R_t values in the same period, as in **Fig. 3d**. **g.** Changes of the average transmissibility across 315 LTLAs during the study period.

Extended Data Figure 1, related to Figure 1. SARS-CoV-2 surveillance sequencing in England between September 2020 and June 2021. a. Local monthly coverage across 315 LTLAs. **b.** Weekly coverage of genomic surveillance sequencing. **c.** Hospitalisation, case and infection fatality rates relative to ONS prevalence.

Extended Data Figure 2: Genomic surveillance model of total incidence and lineage-specific frequencies. **a.** Cubic basis splines (top row) are convolved with the infection to test distribution (row 2 and 3) and used to fit the log incidence in a LTLA and its corresponding derivatives (growth rates; bottom row). **b.** Example incidence (top row), logarithmic incidence with individual convolved basis functions (dashed lines, row 2), growth rate with individual spline basis derivatives (dashed lines, row 3) and resulting (case) reproduction numbers (growth rate per 5.1d) from our approach (GenomicSurveillance) and estimates by EpiEstim⁴⁷, shifted by 10d to approximate a case reproduction number. **c.** The relative frequencies of 62 different lineages are modelled using piecewise multinomial logistic regression. The linear logits are modelled to jump stochastically within 21d prior to first observation to account for the effects of new introductions. Shown are the logits of 5 selected lineages in two different LTLAs.

Extended Data Figure 3: Genomic surveillance model selection. **a.** Model loss in terms of the ELBO objective function and the model hyperparameters α_0 and α_1 (see **Methods**). **b.** Model deviance (calculated as $-2 \times \log$ pointwise predictive density) with respect to the model hyperparameters α_0 and α_1 (see **Methods**). **c.** Mean squared error (MSE) of modelled weekly proportions of highly prevalent lineages with respect to the model parameters α_0 and α_1 (see **Methods**). **d.** Same as in **c**, but for lineages exhibiting low frequencies (VOCs).

Extended Data Figure 4. Spatiotemporal model of 71 SARS-CoV-2 lineages in 315 English LTLAs between September 2020 and June 2021. a. Regional lineage specific relative frequency of lineages contributing more than 50 genomes during the time period shown. Dots denote observed data, lines the fits aggregated to each region. **b.** Same as **a**, but on a log scale. **c.** Same data as in **a**, shown as stacked bar charts. Colors resemble major lineages as indicated and shadings thereof indicate sublineages. **d.** Same fits as in **a**, shown as stacked segments. **e.** Average growth rates for 71 SARS-Cov2 lineages estimated in different regions in England.

Extended Data Figure 5. Relative growth of B.1.177. **a.** Lineage-specific relative frequency data in England, excluding B.1.1.7 and other VOCs/VUIs (Category Other includes: A, A.18, A.20, A.23, A.25, A.27, A.28, B, B.29, B.40, None). Colors resemble major lineages as indicated and shadings thereof indicate sublineages. **b.** Lineage-specific relative frequency data in Denmark, excluding B.1.1.7 and other VOCs/VUIs. Colors resemble major lineages as indicated and shadings thereof indicate sublineages.

Extended Data Figure 6. Genomic diversity of the SARS-CoV-2 epidemic. Shown is the entropy (blue), total number of observed Pango lineages (grey, divided by 4), as well as the proportion of B.1.1.7 (orange, right axis). The sweep of B.1.1.7 causes an intermittent decline of genomic diversity as measured by the entropy.

Extended Data Figure 7. Global phylogenetic trees of selected VOCs/VUIs. English surveillance and other (targeted and quarantine) samples are highlighted respectively orange and red.

Extended Data Figure 8. Global phylogenetic trees of B.1.617 sublineages. a, b and c. English surveillance and other (targeted and quarantine) samples are highlighted respectively orange and red. The trees of B.1.617.1 and B.1.617.2 are rooted. **d.** Number of UK introductions inferred by parsimony (minimum and maximum numbers) and by Thorney BEAST (95% posterior CI) for each VOC.

Referee #1 (Remarks to the Author):

This is a very timely and interesting manuscript. In general, the analysis presented is of very good quality. However, the manuscript gives the impression of three separate analyses of the dataset and that the authors rushed to put the final manuscript.

Thank you for the positive summary of the manuscript. The reviewer raises a valid point that given the pace of the epidemic some parts of the analysis were inconsistent. At the time of the original submission there was still some uncertainty around the nature of the Delta variant, which is why we had chosen to include this only as an addendum. With a little more time and data available now we have addressed these issues and present a coherent analysis covering the period from September 2020 to the end of June 2021.

In order to improve the manuscript, I suggest that the authors select the same date for the final analysis of the dataset and use this date across all figures, as at the moment it is quite confusing. For example, the first part of the manuscript focused on the expansion of B.1.1.7 and ends in March 2021. The second part mentions the replacement of B.1.1.7 with VOCs with 484K, which with hindsight, may not really have happened as the Delta (B.1.617.2) has taken over. The figures are also difficult to compare as they use different data and dates, for example, figures 1, 2 and 3 dates go until March 2021, but Figure 4 dates end up in May 2021 (Figure 4a) and April 2021 (Figure 4b). Figure 5 is even more confusing as different panels (A-E) present data until March, April, and May 2021.

Thank you for highlighting these issues. We have defined a single cut off point of 26.6.2021 and have conducted a single consistent analysis. Accordingly we have updated all Figures of the manuscript. In **Figure 3** we have restricted the date range shown to the period of interest, which were the restrictions in late 2020, when Alpha spread. This is now noted in the legend.

“Figure 3. Growth of B.1.1.7/Alpha and other lineages in relation to lockdown restrictions between November 2020 and March 2021.”

The authors also need to decide the LTLAs that they are discussing in the manuscript. At present, 50 selected LTLAs are presented in Figure 1 but only six LTLAs in Figure 2 and four in LTLAs Figure 3. In addition, given that the LTLAs selected are not the same between Figure 2 and 3, it gives the impression to the reader that the authors may be 'cherry-picking' the LTLAs that fit their hypotheses. I personally do not believe that this is the case but maybe good to select a similar number of LTLAs in Figures 2 and 3 and to try to select the same LTLAs or provide a rationale for the selection of the LTLAs for the figures.

Thank you for raising this. We have reduced the number of example LTLAs in **Figure 2** to 35, selected to uniformly cover England. We chose to do so to provide the reader with a feel for the granularity and offer a glimpse how heterogeneous the pandemic was in time and space. We now ensure that all of the LTLAs shown later in greater detail to illustrate particular periods (the spread of Alpha during restrictions in late 2020 in **Figure 3**; the growth of other VOCs and Delta in the spring of 2021 in **Figure 5**) are also present in the overview of **Figure 2**.

Further, to avoid the impression that the model would break down in other areas, we are providing detailed figures for all 315 LTLAs as a **Supplementary Note**.

The discussion is also very confusing with all of the different endpoints (March, April, May 2021).

Thank you for this comment. We believe that this is now resolved by using a single data set. In order to guide the reader through the manuscript, we have added a clarification:

“Here, we leverage a subset of those data: genomic surveillance generated by the Wellcome Sanger Institute Covid-19 Surveillance Team as part of COG-UK, to characterise the growth rates and geographic spread of different SARS-CoV-2 lineages and reconstruct how newly emerging variants changed the course of the epidemic. We will discuss the key events of the reconstructed epidemic in chronological order.”

The discussion section also lacks a paragraph on the limitation of the analyses of the data.

Thank you for suggesting this. We have added a paragraph with major limitations (stochastic dynamics of growth not accounted for, no model of changing immunity) at the start of the Discussion and refer to a longer discussion of further caveats at the end of the Methods section.

*“Identifying lineages which consistently grew faster than others in each local authority – and thus at the same time, under the same restrictions and in a comparable population – pinpointed a series of variants with elevated transmissibility, in broad agreement with other reports^{10,11,13,15,32}. We note our precise growth rate estimates have a number of limitations. The growth rates of novel and thus rare variants is stochastic due to introductions and local outbreaks. Transmission depends both on the viral variant and the immunity of the host population, which changed from less than 20% to over 90% in the study period³⁹. This will influence the growth rates of VOCs/VUIs with immune evasion capabilities over time. The effect of immunity is currently not modelled, but may become more important in the future as SARS-CoV-2 becomes endemic. Further technical considerations are discussed at the end of the **Methods** section.”*

Further limitations discussed in the Methods section (due to word count limitations in the main text) are:

“Limitations

A main limitation of the model is that the underlying transmission dynamics are deterministic and stochastic growth dynamics are only accounted for in terms of (uncorrelated) overdispersion. For that reason the estimated growth rates may not accurately reflect the viral transmissibility, especially a low prevalence. While the logistic growth assumption is a consistent estimator of the average transmission dynamics, individual outbreaks may deviate from these dynamics and therefore provide unreliable estimates. It is therefore important to assess whether consistent growth patterns in multiple independent areas are observed.

In its current form the model only accounts for a single introduction event per LTLA. While this problem is in part alleviated by the high spatial resolution, which spreads introductions across 315 LTLAs, it is important to investigate whether sustained introductions inflate the observed growth rates, as in the case of the Delta variant or other VOCs and VUIs. This can be achieved by a more detailed phylogeographic assessment and the assessment of monophyletic sublineages.

Furthermore there is no explicit transmission modelled from one LTLA to another. As each introduction is therefore modelled separately, this makes the model conservative in ascertaining elevated transmission as single observed cases across different LTLAs can be explained by their introduction.

The inferred growth rates also cannot identify a particular mechanism which may be caused by higher viral load, longer infectivity or greater susceptibility. Lineages could potentially differ by their inter-generation time, which would lead to a non linear scaling. Here we did not find convincing evidence in incidence data for such effects. in contrast to previous reports 24. However, contact tracing data indicates that the inter-generation time may be shortening for more transmissible lineages such as Delta^{33,61}.

Also lineages, such as Beta, Gamma or Delta differ in their ability to evade prior immunity. As immunity changes over time, this might lead to a differential growth advantage over time. It is therefore advisable to assess whether a growth advantage is constant over periods in which immunity changes considerably.

A further limitation underlies the nature of lineage definition and assignment. The PANGO lineage definition⁵ assigns lineages to geographic clusters, which have by definition expanded, which can induce a certain survivor bias, often followed by winner's curse. Another issue results from the fact that very recent variants may not be classified as a lineage despite having grown, which can inflate the growth rate of ancestral lineages over sublineages.

As the total incidence is modelled based on the total number of positive PCR tests it may be influenced by testing capacity with the total number of tests having approximately tripled between September 2020 and March 2021. This can potentially lead to a time trend in recorded cases and thus baseline R_t values if the access to testing changed, e.g. by too few available tests during high incidence, or changes to the eligibility to test with fewer symptoms intermittently. Generally, the observed incidence was in good agreement with representative cross-sectional estimates from the Office of National Statistics^{62,63}, except for a period of peak incidence from late December 2020 to January 2021 (Extended Data Figure 1d). Values after March 8, 2021 need to be interpreted with caution as pillar 2 PCR testing was supplemented by lateral flow devices, which increased the number of daily tests to more than 1.5 million.

The modelled curves are smoothed over intervals of approximately 7 days using cubic splines, creating a possibility that later time points influence the period of investigation and cause a certain waviness of the R_t value pattern. An alternative parameterization using piecewise linear basis functions per week (i.e., constant R_t values per week) leaves the overall conclusions and extracted parameters broadly unchanged."

In addition, I was surprised to see the conclusion of the manuscript, which is not in line with the results presented or discussed: "The global presence of SARS-CoV-2 in human populations, the existence of human-to-animal and animal-to-animal transmission means it is unlikely that global eradication is possible at least in the short to mid term. Therefore a global perspective on controlling and surveilling the virus is essential. "

We have removed this paragraph. However, we are discussing the consequences of the increased transmissibility of Delta and also the need for global genomic surveillance:

“The 2.4-fold increase in growth rate during the study period as a result of new variants is also likely to have consequences for the future course of the pandemic. If this increase in growth rate was explained solely by higher transmissibility it would raise the basic reproduction number R_0 from a value of around 2.5-3 in the spring of 2020⁴⁰ the range of 6-7 for Delta. This is likely to spur new waves of the epidemic in countries which have so far been able to control the epidemic despite low vaccination rates and may exacerbate the situation elsewhere. Even though the exact herd immunity threshold depends on contact patterns and the distribution of immunity across age groups^{41,42}, it is worth considering that Delta may increase the threshold to values around 0.85. Given current estimates of vaccine efficacy^{34,35,43} this would require nearly 100% vaccination coverage. Even though more than 90% of adults had antibodies against SARS-CoV-2 and close to 70% had received two doses of vaccination, England saw rising Delta variant cases in the first weeks of July 2021. It can thus be expected that other countries with high vaccination coverage are also likely to experience rising cases when restrictions are lifted.

SARS-CoV-2 is likely to continue its evolutionary adaptation process to humans⁴⁴. Thus far variants with considerably higher transmissibility have had strongest positive selection, and swept through England during the 10 months of this investigation. But the possibility that an increasingly immune population may now select for variants with better immune escape highlights the need for continued systematic, and ideally global, genomic surveillance of the virus.”

In summary, my main suggestion is that the author re-write the manuscript with a clear message of how VOCs are replacing each other during the pandemic in England. Many of the analyses will need to be re-done with a unique end date, either on the spring of 2021 or the summer of 2021. Once all the figures relate to each other, we may end up with a superb manuscript that describes how the more transmissible VOCs replaced it other in England and how restrictions affect the epidemic over time.

Thank you for suggesting this. We have redone all analyses as requested and rewritten the manuscript accordingly and focus particularly on the phenomenon of how two much more transmissible variants swept through the country and grew despite restrictions sufficient to contain previous variants.

Referee #2 (Remarks to the Author):

The manuscript by Vöhringer and colleagues documents the genomic epidemiology of SARS-CoV-2 in England during a critical time period of the COVID-19 pandemic. The authors elegantly demonstrate how lineage information can be integrated with incidence data in a spatio-temporal surveillance model, delivering insights with a remarkable resolution, including estimates of lineage-specific growth rates. Although the general picture of lineage turnover and its impact on transmission dynamics was already evident thanks to near real-time surveillance efforts, this study also takes advantage of the dense sampling to uncover more detailed dynamics 'under the surface' such as the low-level persistence or slow rise of variants with the E484K spike mutation from December to April. While various studies have already highlighted the tremendous efforts by the COG-UK and its exemplary role in SARS-COV-2 surveillance, I believe the current study may set yet another standard for modern large-scale genomic epidemiology. The authors should also be commended for the short turnaround time in putting together the current study.

We thank the reviewer for this very positive overall summary.

At the core of this study stands a state-of-the-art spatio-temporal surveillance model that integrates incidence and genomic data. The Authors adequately outline the limitations of the model but ask the reader to visually assess model fit (in Figure 2 b & c, and Supplementary Figure 3). While I do not wish to question a good fit, in particular at the level of larger regions, a more formal assessment would be desirable. I am wondering for example whether a cross-validation approach could be applicable in this setting.

The reviewer raises an important issue, which we have addressed in the revision. We are providing different metrics for the quality of fit (MSE, deviance, ELBO) to provide a quantitative understanding of the model's current accuracy and also in order to justify certain parameter choices. These findings are shown in **Extended Data Figure 3**. We are further providing a cross-validation of the inferred growth rates by fitting models only to LTLAs from each of 6 English regions. The resulting growth rates are in good agreement with one another. The results are shown in **Extended Data Figure 4e**.

Given the fast-evolving epidemic dynamics and the communication of findings from near real-time surveillance, it is likely that the recent rapid rise of B.1.617.2 may receive most of the attention. The Authors present a growth estimate for B.1.617.2 of about 37% in excess of that of B.1.1.7, but remain nuanced in their interpretation by pointing at different factors that could be responsible for this. Data that has accumulated more recently may help to shed further light on this. I would not want to insist to update this analysis as manuscripts will always lag behind the latest available data. However, the authors may have anticipated this and closely followed up the situation, in which case an update may be not be too cumbersome.

Thank you for highlighting the importance of the analyses related to the Delta variant. As suggested, we are now discussing more recent finding related to its epidemiology, transmissibility increase and immune evasion capabilities. Further and as discussed further above we are presenting a coherent analysis until the end of June 2020, which covers Delta's sweep in the UK.

Minor text errors:

Nevertheless B.1.1.7 nevertheless

Figure 5, caption: ...in two selected LTLAs -> in four selected LTLAs?

Methods, Limitations: especially a low prevalence -> especially at low prevalence?

These have been corrected. Thank you.

Referee #3 (Remarks to the Author):

A. Summary of key results

The authors do a very nice job of providing a summary of their key results via four "chapters" in their Discussion. This includes that: 1) the intro of B.1.177 became dominant with a peak in October 2020; 2) B.1.1.7 became dominant via a selective sweep of other lineages; 3) A December 2020 was needed to slow B.1.1.7 which also contracted other lineages that were eliminated in Spring 2021; 4) the E484K mutation in the Spike protein were introduced repeatedly from December to April 2021.

B. Originality and significance

The work is original and significant and leverages the strength of the UK Genomic Consortium and includes a sequencing rate of over 5% of cases.

Thank you for this positive assessment.

C. Data & methodology

The work includes mathematical modeling and phylogenetic analysis. It was well done but some concerns:

1) It was surprising that no vaccination data was considered even in the later months of the models and in May 2021.

We do agree that the role of vaccination is critical in assessing the dynamics of the epidemic and also in relation to particular variants. This is probably as important as ever in the context of wide-spread natural and vaccine-derived immunity. Unfortunately, it is not easy to directly incorporate this into our modeling framework as it would require keeping track of vaccination or prior infection status of every case and sample and of the local population. This is possible in theory, but such data are not readily available to us, unfortunately.

Since we believe that the reviewer has raised an important issue, we have further clarified the role of immunity and immune evasion in the text.

"Delta's sustained domestic growth and international spread³² relative to the Alpha lineage are first evidence of a biological growth advantage. Causes appear to be a combination of increased transmissibility and immune evasion. Evidence for higher transmissibility are the high rates of spread in younger, unvaccinated age groups, reports of elevated secondary attack rates¹⁷ and a higher viral load³³. Further, vaccine efficacy against infection by Delta is diminished, depending on the type of vaccine^{34,35} and reinfection is more frequent³⁶, both supported by experimental work demonstrating reduced antibody neutralisation of Delta by vaccine derived and convalescent sera^{37,38}."

We also highlight that this is a limitation of our inference in a new limitations paragraph at the beginning of the Discussion.

"Transmission depends both on the viral variant and the immunity of the host population, which changed from less than 20% to over 90% in the study period. This will influence the growth rates of

VOCs/VUIs with immune evasion capabilities over time. The effect of immunity is currently not modelled, but may become more important in the future as SARS-CoV-2 becomes endemic.”

2) For phylogenetics, were duplicate weekly sequences by LTLA removed?

Thank you for raising this. We did not remove duplicate sequences per LTLA. Duplicate sequences from different individuals are often found in SARS-CoV-2 given its rate of evolution, and represent an important signal to detect rapidly spreading variants. We now clarify this in the second paragraph of the “Phylogeographic analyses” section of the Methods: “Multiple occurrences of identical sequences were counted as separate cases, since this helps us identify rapid SARS-CoV-2 spread”.

3) More description on the parsimony approach for introductions. Why was Bayesian phylogenetics with ancestral state reconstruction not considered? There are new approaches for analyzing larger datasets (https://beast.community/thorney_beast).

Thank you for this comment. We have largely extended the Methods description of our parsimony approach:

“Our approach works by traversing the maximum likelihood tree starting from the terminal nodes and ending at the root (postorder traversal). Here, we define a “UK clade” as a maximal subtree of the total phylogeny for which all terminal nodes are from the UK, all internal nodes are inferred to be from the UK, and at least one terminal node is a UK surveillance sample; the size of a UK clade is defined as the number of UK surveillance samples in it. At each node, using values already calculated for all children nodes (possibly more than two children in the case of a multifurcation), we calculate the following quantities: i) the maximum and minimum number of possible descendant UK clades of the current node, over the space of possible parsimonious migration histories, and conditional on the current node being UK or non-UK; ii) the number of migration events compatible with a parsimonious migration history in the subtree below the current node, and conditional on the current node being UK or non-UK; iii) the size so far of the UK clade the current node is part of, conditional on it being UK; iv) A sample of UK clade sizes for the subtree below the node. To calculate these quantities, for each internal node, and conditional on each possible node state (UK or non-UK), we consider the possible scenarios of having 0 or 1 migration event between the internal node and its children nodes (migration histories with more than 1 migration event between the node and its children are surely not parsimonious in our analysis and can be ignored).”

And we have expanded the justification:

“Parsimony is expected to represent a good approximation in the context of SARS-CoV-2, due to the shortness (both in time and substitutions) of the phylogenetic branches considered^{53,54}. The main advantage of our approach is that, thanks to the dynamic programming implementation, it’s more computationally efficient than Bayesian alternatives, as the most computationally demanding step is the inference of the maximum likelihood phylogenetic tree. This allows us to infer plausible ranges for numbers of introduction events for large datasets and to quickly update our analyses as new sequences become available. The other advantage of this approach is that it allows us to easily customize the analysis and to focus on inferred UK introductions that result in at least one UK surveillance sample, while still making use of non-surveillance UK samples to inform the inferred phylogenetic tree and migration history.”

Moreover, we have now additionally performed some of the phylogeographic analyses using thorney BEAST. Overall, BEAST infers more introductions than parsimony (as expected of a Bayesian versus a parsimony approach) but overall results agree between the two methods regarding which VOCs underwent most introduction to the UK. This is shown in **Extended Data Figure 8d**.

We included the following details of the new Thorney BEAST analyses in the manuscript:

"To confirm the results of our analyses based on parsimony, we have also used the new Bayesian phylogenetic approach Thorney BEAST¹⁶ (https://beast.community/thorney_beast) for VOCs for which it was computationally feasible, that is, excluding B.1.351. For each VOC, we used in Thorney BEAST the same topology inferred with FastTree2 as for our parsimony analysis; in addition, we used treetime⁵⁶ 0.8.2 to estimate a timed tree and branch divergences for use in Thorney BEAST. We used a 2-state ("UK" and "non-UK") migration model of migration to infer introductions into the UK, but again, only counted, from the posterior sample trees, UK clades with at least one UK surveillance sample. We used a Skygrid⁵⁷ tree coalescent prior with 6 time intervals."

4) What non-UK countries were included? That should be clearer in the main manuscript.

We have included all countries with samples provided to GISAID in our analysis. This is now clarified in the first paragraph of the "Phylogeographic analyses" in Methods: *"we investigated VOC genome sequences from GISAID <https://www.gisaid.org/> available from any country"*.

D. Appropriate use of statistics and treatment of uncertainties

No concerns

Thank you

E. Conclusions: robustness, validity, reliability

Discussion section needs improvement.

1) No consideration of other literature given the implications of the study findings.

2) No Limitations paragraph

Thank you for these two suggestions. We have added the following limitations paragraph and also refer to a longer discussion at the end of the Methods section:

"Identifying lineages which consistently grew faster than others in each local authority – and thus at the same time, under the same restrictions and in a comparable population – pinpointed a series of variants with elevated transmissibility, in broad agreement with other reports^{10,11,13,15,32}. We note our precise growth rate estimates have a number of limitations. The growth rates of novel and thus rare variants is stochastic due to introductions and local outbreaks. Transmission depends both on the viral variant and the immunity of the host population, which changed from less than 20% to over 90% in the study period³⁹. This will influence the growth rates of VOCs/VUIs with immune evasion capabilities over time. The effect of immunity is currently not modelled, but may become more important in the future as SARS-CoV-2 becomes endemic. Further technical considerations are discussed at the end of the Methods section."

The Limitation paragraph at the end of the Methods section is

“Limitations

A main limitation of the model is that the underlying transmission dynamics are deterministic and stochastic growth dynamics are only accounted for in terms of (uncorrelated) overdispersion. For that reason the estimated growth rates may not accurately reflect the viral transmissibility, especially a low prevalence. While the logistic growth assumption is a consistent estimator of the average transmission dynamics, individual outbreaks may deviate from these dynamics and therefore provide unreliable estimates. It is therefore important to assess whether consistent growth patterns in multiple independent areas are observed.

In its current form the model only accounts for a single introduction event per LTLA. While this problem is in part alleviated by the high spatial resolution, which spreads introductions across 315 LTLAs, it is important to investigate whether sustained introductions inflate the observed growth rates, as in the case of the Delta variant or other VOCs and VUIs. This can be achieved by a more detailed phylogeographic assessment and the assessment of monophyletic sublineages.

Furthermore there is no explicit transmission modelled from one LTLA to another. As each introduction is therefore modelled separately, this makes the model conservative in ascertaining elevated transmission as single observed cases across different LTLAs can be explained by their introduction.

The inferred growth rates also cannot identify a particular mechanism which may be caused by higher viral load, longer infectivity or greater susceptibility. Lineages could potentially differ by their inter-generation time, which would lead to a non linear scaling. Here we did not find convincing evidence in incidence data for such effects. in contrast to previous reports 24. However, contact tracing data indicates that the inter-generation time may be shortening for more transmissible lineages such as Delta^{33,61}.

Also lineages, such as Beta, Gamma or Delta differ in their ability to evade prior immunity. As immunity changes over time, this might lead to a differential growth advantage over time. It is therefore advisable to assess whether a growth advantage is constant over periods in which immunity changes considerably.

A further limitation underlies the nature of lineage definition and assignment. The PANGO lineage definition⁵ assigns lineages to geographic clusters, which have by definition expanded, which can induce a certain survivor bias, often followed by winner’s curse. Another issue results from the fact that very recent variants may not be classified as a lineage despite having grown, which can inflate the growth rate of ancestral lineages over sublineages.

As the total incidence is modelled based on the total number of positive PCR tests it may be influenced by testing capacity with the total number of tests having approximately tripled between September 2020 and March 2021. This can potentially lead to a time trend in recorded cases and thus baseline Rt values if the access to testing changed, e.g. by too few available tests during high incidence, or changes to the eligibility to test with fewer symptoms intermittently. Generally, the observed incidence was in good agreement with representative cross-sectional estimates from the Office of National Statistics^{62,63}, except for a period of peak incidence from late December 2020 to January 2021 (Extended Data Figure 1d). Values after March 8, 2021 need to be interpreted with caution as pillar 2 PCR testing was supplemented by lateral flow devices, which increased the number of daily tests to more than 1.5 million.

The modelled curves are smoothed over intervals of approximately 7 days using cubic splines, creating a possibility that later time points influence the period of investigation and cause a certain waviness of the R_t value pattern. An alternative parameterization using piecewise linear basis functions per week (i.e., constant R_t values per week) leaves the overall conclusions and extracted parameters broadly unchanged.”

3) Now that we have these results, what do these results suggest for UK to best protect themselves?

We believe that there are basically two key takeaways from our analysis, which apply both to the UK and probably also to the rest of the world:

1. There have been two sweeps in the UK caused by more transmissible variants, which rendered previously appropriate countermeasures ineffective. This shows that rapid genomic surveillance is important to inform policy. Further, the fact that the Delta lineage emerged in India, but was only fully characterised once it had spread to the UK and other countries, shows that such surveillance should be global.
2. Transmissibility more than doubled with Delta. This implies that reaching herd immunity with current vaccines alone requires almost complete vaccination coverage. The alternative is a massive exit wave once restrictions are lifted.

We have included these thoughts into the rewritten Discussion, which concludes:

“The 2.4-fold increase in growth rate during the study period as a result of new variants is also likely to have consequences for the future course of the pandemic. If this increase in growth rate was explained solely by higher transmissibility it would raise the basic reproduction number R_0 from a value of around 2.5-3 in the spring of 2020⁴⁰ the range of 6-7 for Delta. This is likely to spur new waves of the epidemic in countries which have so far been able to control the epidemic despite low vaccination rates and may exacerbate the situation elsewhere. Even though the exact herd immunity threshold depends on contact patterns and the distribution of immunity across age groups^{41,42}, it is worth considering that Delta may increase the threshold to values around 0.85. Given current estimates of vaccine efficacy^{34,35,43} this would require nearly 100% vaccination coverage. Even though more than 90% of adults had antibodies against SARS-CoV-239 and close to 70% had received two doses of vaccination, England saw rising Delta variant cases in the first weeks of July 2021. It can thus be expected that other countries with high vaccination coverage are also likely to experience rising cases when restrictions are lifted.

SARS-CoV-2 is likely to continue its evolutionary adaptation process to humans⁴⁴. Thus far variants with considerably higher transmissibility have had strongest positive selection, and swept through England during the 10 months of this investigation. But the possibility that an increasingly immune population may now select for variants with better immune escape highlights the need for continued systematic, and ideally global, genomic surveillance of the virus.”

F. Suggested improvements

In addition to the concerns listed above.

1) Suggest adding Greek letters in addition to Pangolin lineages to be consistent with new standards.

We have added greek letters as requested and use these preferentially for WHO variants of concern, as this appears to be the new standard.

2) Better explanation of the scientific premise (a priori hypothesis) before starting the study. What were the key questions that needed to be answered.

We have added the following short sentence at the beginning of the Results to clarify the aim of the investigation:

“Here, we leverage a subset of those data: genomic surveillance generated by the Wellcome Sanger Institute Covid-19 Surveillance Team as part of COG-UK, to characterise the growth rates and geographic spread of different SARS-CoV-2 lineages and reconstruct how newly emerging variants changed the course of the epidemic. We will discuss the key events of the reconstructed epidemic in chronological order.”

3) Explain the choice for the study period.

As also requested by the other two reviewers, we have extended the analysis range to the period from September 2020 to June 2021, which covers the sweeps of the Alpha and Delta variants in England.

The preceding first wave of the epidemic in England from March to June 2020, during which testing and surveillance sequencing was not as widespread yet, has been covered by previous studies, such as:

du Plessis, L. et al. Establishment and lineage dynamics of the SARS-CoV-2 epidemic in the UK. *Science* 371, 708–712 (2021)

Volz, E. et al. Evaluating the Effects of SARS-CoV-2 Spike Mutation D614G on Transmissibility and Pathogenicity. *Cell* 184, 64–75.e11 (2021).

4) Figure 1C. Suggest a color-blind check of the colors used for the heat map.

Thank you for this suggestion. We have investigated some guidance and believe that using a single color gradient per row would also help understand the geographic distribution of each selected lineage.

5) Figure 2B-C. Relative frequencies really small and hard to visualize. Removing panel d might help a little or making one of them separate.

To make this overview Figure, which was thought to provide the reader with a feel for the granularity of the data and diverse nature of the epidemic, less distracting, we have reduced the number of areas shown to 30. These are now in line with the larger versions shown in Figures 3 and 5. As also

discussed in response to reviewer 2 and 3 we are showing the analysis of all LTLAs in a **Supplementary Note**.

6) **"Furthermore, at low frequency growth can be stochastic..." remove "at"**

This has been corrected.

G. References

Appropriate

Thank you

H. Clarity and context

Main concern was with Discussion. Already addressed.

Reviewer Reports on the First Revision:

Referee #1 (Remarks to the Author):

The manuscript has improved and now presents the results in a consistent spatial and temporal manner. For example, the end date of the analyses is now consistent as well the selection of the English Lower Tier Local Authorities (LTLAs). In addition, a limitation section was added in the discussion and material and methods. I have no further comments and look forward to seeing it published.

Referee #2 (Remarks to the Author):

The Authors have adequately addressed my comments and I have no further concerns. This is an impressive piece of work.

Referee #3 (Remarks to the Author):

The authors have done a very rigorous response to my critiques including implementation of "pre thorney" Bayesian MCMC analysis in BEAST to compare introductions with their original parsimonious approach. They have also incorporated the Greek letters as labels for the VoCs to be consistent with current nomenclature. Their expanses of the Discussion based on known limitations is also appropriate.

Author Rebuttals to First Revision:

Point by point replies

Referees' comments:

Referee #1 (Remarks to the Author):

The manuscript has improved and now presents the results in a consistent spatial and temporal manner. For example, the end date of the analyses is now consistent as well the selection of the English Lower Tier Local Authorities (LTLAs). In addition, a limitation section was added in the discussion and material and methods. I have no further comments and look forward to seeing it published.

Referee #2 (Remarks to the Author):

The Authors have adequately addressed my comments and I have no further concerns. This is an impressive piece of work.

Referee #3 (Remarks to the Author):

The authors have done a very rigorous response to my critiques including implementation of "pre thorny" Bayesian MCMC analysis in BEAST to compare introductions with their original parsimonious approach. They have also incorporated the Greek letters as labels for the VoCs to be consistent with current nomenclature. Their expanses of the Discussion based on known limitations is also appropriate.

We are pleased to hear that the reviewers are satisfied with our revisions.